# CaTs and DAGs: Integrating Directed Acyclic Graphs with Transformers for Causally Constrained Predictions

**Matthew J. Vowels**
Kivira Health.
The Sense Innovation and Research Center, CHUV, Lausanne, Switzerland.
Centre for Vision, Speech and Signal Processing, University of Surrey, U.K.
`matt@kivira.health`

**Mathieu Rochat**
Institute of Psychology, University of Lausanne, Switzerland.
`mathieu.rochat@unil.ch`

**Sina Akbari**
Statistical Laboratory, University of Cambridge, U.K.
`sa2385@cam.ac.uk`

## Abstract

Artificial Neural Networks (ANNs), including fully-connected networks and transformers, are highly flexible and powerful function approximators, widely applied in fields like computer vision and natural language processing. However, their inability to inherently respect causal structures can limit their robustness, making them vulnerable to covariate shift and difficult to interpret/explain. This poses significant challenges for their reliability in real-world applications. In this paper, we introduce Causal Transformers (CaTs), a general model class designed to operate under predefined causal constraints, as specified by a Directed Acyclic Graph (DAG). CaTs retain the powerful function approximation abilities of traditional neural networks while adhering to the underlying structural constraints, improving robustness, reliability, and interpretability at inference time. This approach opens new avenues for deploying neural networks in more demanding, real-world scenarios where robustness and explainability is critical.

## 1 Introduction

Machine learning has long focused on discovering powerful function approximation techniques for mapping inputs to outputs. Two of the most widely used approaches are the fully-connected neural network and the transformer (Vaswani et al., 2017), both of which are Artificial Neural Networks (ANNs). These methods have achieved broad success in computer vision (Dosovitskiy et al., 2021), audio (Radford et al., 2023), reinforcement learning (Vinyals et al., 2019), natural language processing (Devlin et al., 2019; Brown et al., 2020), and other domains (J. et al., 2021). Despite their efficacy, ANNs rely primarily on statistical, data-driven mechanisms and do not typically incorporate prior knowledge about the underlying Data Generating Process (DGP).

Models that fail to respect the DGP often show sensitivity to covariate shift (Arjovsky et al., 2020; Rojas-Carulla et al., 2018; de Haan et al., 2019). They may exploit non-causal associations in training data - relationships that lack mechanistic meaning - and underperform when such correlations change or are irrelevant to the outcome (see Fig. 1). A well-known example is classifying camels versus cows: training data often confound animals with their backgrounds (camels in sandy regions, cows in green pastures) (Arjovsky et al., 2020). A non-causal model may rely on the background rather than the animal itself, leading to errors such as misclassifying a camel in a grassy field. In

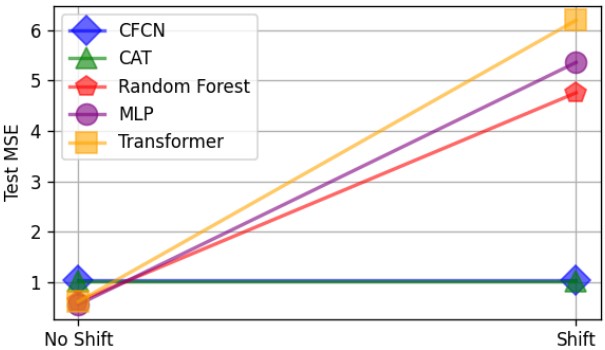

Figure 1: Illustrating the lack of covariate shift robustness associated with conventional machine learning models like random forests (Breiman, 2001), multilayer perceptrons (MLPs), and transformers (Vaswani et al., 2017), compared with our CaT and CFCN.

general, non-causal ANNs entangle distinct factors, reducing robustness and making predictions highly sensitive to irrelevant features.

A central goal of science, however, is to understand causality (Pearl, 2009; Hernan, 2018; Rohrer, 2018; Scholkopf, 2019; Peters et al., 2017). Causal *inference* seeks to estimate the effect of an intervention (e.g., taking a drug) even in the absence of randomized data, a task made difficult by confounding. For instance, comparing outcomes between patients who select surgery versus medication is biased by age - older patients are more likely to choose medication and less likely to recover. Without causal structure, models typically yield biased effect estimates (Cadei et al., 2024; Vowels, 2023a; Pearl, 2009). To isolate true causal effects from spurious associations, explicit structural constraints are required, beyond what "purely statistical" approaches in machine learning can provide.

Our main contribution is a modification to the transformer architecture that constrains interactions according to a Directed Acyclic Graph (DAG) representing causal structure. We call this the *Causal Transformer* (CaT). For comparison, we also extend the masking idea from MADE (Germain et al., 2015) to create DAG-constrained fully connected networks, which we term *Causal Fully Connected Networks* (CFCNs). Such constraints reduce sensitivity to covariate shift and broaden applicability across causal domains including medical imaging (Castro et al., 2019), policy evaluation (Kreif & DiazOrdaz, 2019), medicine (Petersen et al., 2017; Deaton & Cartwright, 2018), advertising (Bottou et al., 2013), and social science (Grosz et al., 2020; Vowels, 2023a; Rohrer, 2018).

In summary, CaT[1] extends transformers to respect causal knowledge, enabling their application across empirical domains. CaTs can operate on inputs of arbitrary dimensionality, from tabular variables to high-dimensional multimodal embeddings. The goal of this work is not to provide state-of-the-art performance on specific benchmarks, but to propose a general modeling framework that integrates structural inductive biases into a popular neural architecture. This integration enhances interpretability, fairness (Locatello et al., 2019a), and data efficiency (Vowels, 2023b), while promoting robustness to distributional changes (Arjovsky et al., 2020). These models are intended as foundations for further innovation, allowing researchers to adapt and extend them for domain-specific applications where causal structure is essential.

## 2    RELATED WORK

The design of effective machine learning models often relies on *inductive bias*. Such bias can be introduced in two main ways: (a) A *structural* bias constrains possible interactions between variables, dimensions, or inputs, for example to disentangle generative factors (Locatello et al., 2019b; Higgins et al., 2017; Burgess et al., 2018; Yang et al., 2020; Bengio et al., 2019) or to estimate

---

[1]Code for CaTs and the experiments can be found here: `https://github.com/matthewvowels1/Causal_Transformer`

the effect of an intervention (Shalit et al., 2017; Louizos et al., 2017b). (b) A *functional* bias restricts the class of functions that can be learned, such as convolution for vision tasks (LeCun & Bengio, 1998), regularization for smoothness (Goyal & Bengio, 2022), or translation equivariance (Fuchs et al., 2020). Functional biases also project inputs into specialized spaces, e.g. for registration in medical images (Friedman et al., 2023) or DAG-based emotion prediction (Shen et al., 2021).

Our work falls under the first category. Structural inductive bias has been integrated into neural models for causal inference via multi-task deep networks (Xia et al., 2021; Kocaoglu et al., 2018; Shalit et al., 2017; Shi et al., 2019), Variational Autoencoders (Kingma & Welling, 2014; Louizos et al., 2017a; Zhang et al., 2021; Vowels et al., 2021b), and variants with adversarial (Yoon et al., 2018) or semi-parametric objectives (Shi et al., 2019; Vowels et al., 2023b). Beyond neural networks, Structural Equation Modeling (Kline, 2005) and non-linear extensions (Kelava et al., 2014; Umbach et al., 2017; van Kesteren & Oberski, 2022) remain popular for causal effect estimation, with the added benefit of statistical inference. Unbiased estimators can also be obtained by combining predictive models such as MLPs or Random Forests (Breiman, 2001) with careful predictor selection based on the backdoor criterion (Pearl, 2009).

Most causal inference methods target tabular data, but structural bias has also been applied in vision tasks (Vowels et al., 2021a; Krishnan et al., 2015; Chen et al., 2019; Li & Mandt, 2018), where knowledge of changing versus static factors (e.g. pose vs. identity) helps disentanglement. Our CaT approach can similarly exploit multidimensional embeddings. Indeed, (Vafa et al., 2024) show that missing inductive biases leave generative transformers fragile, highlighting the need for explicit constraints.

Attempts to make transformers "causal" have often focused on autoregressive masking, which restricts attention to past tokens (Vaswani et al., 2017; Yin et al., 2024; Yenduri et al., 2024). This constraint resembles Granger causality (Granger, 1980) and remains weak: it cannot enforce high-level disentanglement, causal inference, or robustness to covariate shift. More targeted work includes (Melnychuk et al., 2022), who use separate transformers for treatment, covariates, and outcomes, but this requires grouping variables by estimand. By contrast, CaT applies to any causal effect implied by a DAG without subnetworks.

Other DAG–transformer combinations pursue different goals. (Luo et al., 2023) allow bidirectional links (violating causal ordering) for outcome prediction, while (Gagrani et al., 2022) predict topological ordering itself. In general, while functional inductive biases (e.g. convolution, weight decay) are common, neural networks rarely enforce structural constraints. Designing such models is challenging because ANNs usually facilitate maximal, not restricted, interactions.

The most comparable work to our baseline (CFCN) includes MADE (Germain et al., 2015), Graphical Normalizing Flows (Wehenkel & Louppe, 2021), and Structured Neural Networks (Chen et al., 2023). MADE masks weights so each output depends only on earlier inputs in an autoregressive order. For example, with variables $A, B, C$ in an autoregressive structure, masks enforce $\hat{C}|(B, A)$, $\hat{B}|A$, and $\hat{A}|\varnothing$. Wehenkel & Louppe (2021) combine normalizing flows with DAGs, while Chen et al. (2023) express adjacency as a binary factorization optimized across layers. In contrast, our approach applies masks directly from the DAG and adapts them to layer shapes, yielding a simple and comparable architectural baseline.

## 3 METHODS

### 3.1 PROBLEM SETUP AND BACKGROUND

In this work we address two main problem setups. We consider dataset $\mathcal{D} = \{[\mathbf{Y}_i, \mathbf{D}_i, \mathbf{L}_{\text{set},i}]\}_{i=1}^N$, whereby $N$ is the sample size, and treatment $\mathbf{D}$, outcome $\mathbf{Y}$, and covariates $\mathbf{L}_{\text{set}} = \{\mathbf{L}^1, ..., \mathbf{L}^{|\mathbf{L}_{\text{set}}|}\}$ are multidimensional embeddings (possibly of different dimensionality). CaT is concerned with this setup, but also functions in the case where all variables are univariate. CFCN, our architectural baseline, functions in the univariate case. We make no assumptions about the parametric or distributional form of each uni- or multi-dimensional variable. We use $B$ to indicate batch size, $|Z|$ to indicate the number of nodes in the DAG, and, $C$ to indicate the feature dimensionality for each variable.

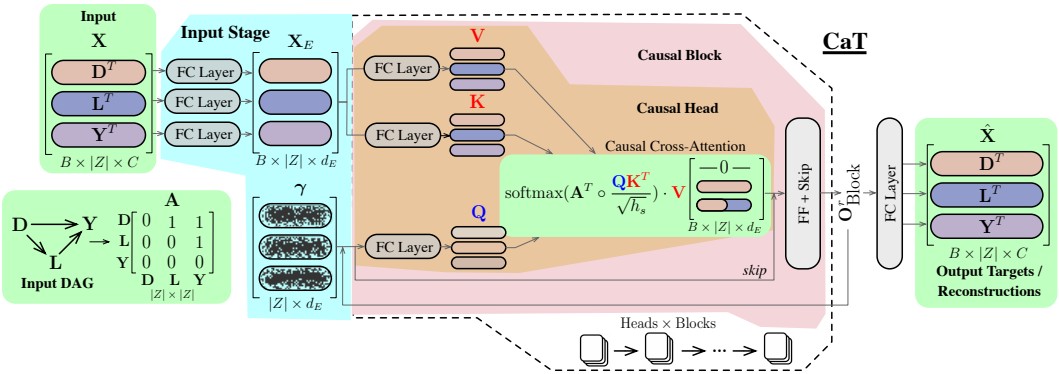

Figure 2: A top-level depiction of how the masking and routing is applied in CaT to an input with a batch $B$ of $|Z| = 3 \times C$-dimensional embeddings and a corresponding causal DAG. The input parameter $\boldsymbol{\gamma}$, which is initially random, is recursively fed as an input to the causal Heads, 'extracting' information from the embeddings of the input via the causal cross-attention operation, according to the constraints imposed by the adjacency matrix $\mathbf{A}$. Best viewed in color. See main text for details.

In both scenarios we are concerned with building a model encoding a set of conditional independencies encoded by a Directed Acyclic Graph (DAG) $\mathcal{G}$. The DAG represents the factorization of a joint distribution $\mathcal{P}(\mathbf{Y}, \mathbf{D}, \mathbf{L}_{\text{set}})$ of all variables using $|Z|$ corresponding nodes $z \in Z$ and edges $(j, k) \in \mathcal{E}$, where $(j, k)$ indicates the presence of a directed edge from node $j$ to node $k$. The acyclicity indicates the absence of any cyclic paths (such that there are no directed paths between a node and itself). DAGs are a popular tool for representing causal structures or DGPs. For instance, the DAG $\mathbf{D} \rightarrow \mathbf{M} \rightarrow \mathbf{Y}$ tells us that $\mathbf{D}$ causes variable $\mathbf{M}$ which itself is a mediator between $\mathbf{D}$ and $\mathbf{Y}$ where $\mathbf{Y}$ here would be the outcome of the causal chain. DAGs can be represented as a square adjacency matrix. With each row representing a possible parent (i.e. direct causal predecessor), and each column representing a possible child (i.e., a variable which is affected by its parent). The adjacency matrix representation will be a key component in enforcing the causal constraints in our CFCNs and CaTs.

The end goal is to be able to reason causally, under the constraints established by the DAG. One associated task is the estimation of the average treatment effect $\tau$, which can be expressed as follows:

$$
\begin{aligned}
\boldsymbol{\tau} :=& \mathbb{E}_{\mathbf{Y} \sim P(\mathbf{Y}|do(\mathbf{D=1}))}[\mathbf{Y}] - \mathbb{E}_{\mathbf{Y} \sim P(\mathbf{Y}|do(\mathbf{D=0}))}[\mathbf{Y}] \\
=& \mathbb{E}_{\mathbf{L}_{\text{set}} \sim \mathcal{P}}\left[\mathbb{E}_{\mathbf{Y} \sim P(\mathbf{Y}|do(\mathbf{D=1}), \mathbf{L}_{\text{set}})}[\mathbf{Y}] - \mathbb{E}_{\mathbf{Y} \sim P(\mathbf{Y}|do(\mathbf{D=0}), \mathbf{L}_{\text{set}})}[\mathbf{Y}]\right].
\end{aligned}
\tag{1}
$$

Here, the $do$ operator (Pearl, 2009) denotes an intervention, simulating the case whereby $\mathbf{D} = \mathbf{0}$ and $\mathbf{D} = \mathbf{1}$, even though in the joint distribution $\mathcal{P}$, $\mathbf{D}$ takes on values according to the underlying DGP. This quantity is therefore hypothetical, but can be estimated unbiasedly under the right conditions, and these conditions can usually be derived from the DAG itself (Pearl, 2009; Peters et al., 2017). In practice, causal discovery techniques, domain knowledge and inductive biases can be used for the specification of a DAG, although in this work we assume the DAG is known *a priori*. One of the key advantages of models that respect the underlying causal structure (or, more loosely, respect some underlying properties of the DGP) is that predictions are less sensitive to distributional shift. Conventional machine learning models which leverage all available correlations make predictions which fluctuate according to changes in spurious or confounding factors (Arjovsky et al., 2020; Vowels, 2022). Causal models, on the other hand, may have less predictive power under a stable joint distribution, but maintain this predictive power under covariate shift precisely because they leverage only those relationships which are invariant under such shift.

*Importantly*, we note that the success of causal inference rests on a number of strong and often untestable assumptions known as ignorability, positivity, and the stable unit treatment value assumption, each of which we describe further in Sec.S8. Interested readers are directed to surveys and introductions by (Vowels et al., 2022; Vowels, 2024; Glymour et al., 2019; Pearl & Mackenzie, 2018; Peters et al., 2017; Scholkopf, 2019) for more information on causality, and Sec. S8 for further preliminaries.

Note that if we relax the requirement for unbiased estimates for causal effects, one can use the DAG to loosely constrain the model according to relevant factors and to make predictions for arbitrary interventions (more on this below). As such, CaTs are not *only* useful for causal inference, but also as models which are robust to covariate shift in proportion to the strength of the constraints imposed via the DAG, and the coherence between this DAG and the true, underlying DGP.

## 3.2 CAUSAL TRANSFORMERS (CATS)

One key limitation of existing ANN-based causal approaches (Vowels et al., 2021b; Shi et al., 2019; Louizos et al., 2017a; Shalit et al., 2017; Yoon et al., 2018) and related methods (van der Laan et al., 2007; Wager & Athey, 2018; Vowels, 2023c) is that they treat each variable as a single value, restricting inference to a uni-dimensional setting. This prevents direct use of high-dimensional embeddings such as skeleton joint sets, voice embeddings, or multimodal features from images and text. In contrast, our transformer-based approach views the input as a sequence of $d$-dimensional embeddings, making it applicable to both tabular data (with scalar variables) and complex multimodal data. This property is particularly useful in applied fields like psychology and medicine, where researchers often rely on multi-item scales or multidimensional outcomes (Hilpert et al., 2019; Bulling et al., 2023). For example, the NEO Personality Inventory comprises 240 items (Costa & McCrae, 2008), typically averaged into five trait scores. With CaTs, each subscale can instead be represented as a vector of items, preserving information that would otherwise be lost.

For a brief review of self- and cross-attention mechanisms in typical transformers, please see Sec. S5. A top-level depiction of the core change to the transformer attention architecture is shown in Fig. 2. The complete process for the single causally-masked cross-attention head of CaT is given below. We provide a proof that the constraints implied by the DAG are retained through the proposed architecture in S13.

First, we embed the input $\mathbf{X}$ from shape $(B \times |Z| \times C)$ into $\mathbf{X}_E$ with dimensionality $(B \times |Z| \times d_E)$. Crucially, this is done using $|Z|$ independent linear layers. It is also important that $d_E$ is at least as large as $C$ and greater than 1. In practice, we find that even if $C = d_E$, but $d_E = 1$, the network struggles to disentangle the contributions from the separate variables for all but trivial cases. This stage can also be used to unify the dimensionality of all input embeddings in the case where the different variables do not have the same dimensionality (this may be the case with multimodal data, for instance).

Next, we randomly initialize a learnable embedding $\boldsymbol{\gamma}$, which is of size $|Z| \times d_E$. This parameter is combined using cross-attention, with the embedded input $\mathbf{X}_E$ throughout the network and according to the interactions implied by the DAG. For cross-attention, the keys, queries and values are computed as follows:

$$\begin{aligned} \mathbf{K} &= \mathbf{X}_E \mathbf{W}_K + \mathbf{b}_K, \\ \mathbf{Q} &= \boldsymbol{\gamma} \mathbf{W}_Q + \mathbf{b}_Q, \\ \mathbf{V} &= \mathbf{X}_E \mathbf{W}_V + \mathbf{b}_V. \end{aligned} \tag{2}$$

Importantly, note that whilst the keys and values are functions of the embedded input $\mathbf{X}_E$, the query is a function of the learnable embedding $\boldsymbol{\gamma}$ in the first block, and a function of the output of the previous block for subsequent blocks (see Eq. 6).

We then mask the attention between $\mathbf{Q}$ (which is a function of $\boldsymbol{\gamma}$) and $\mathbf{K}^\top$ (which is a function of $\mathbf{X}_E$). The mask is applied via a Hadamard product (indicated by $\circ$) between the transposed, topologically sorted, adjacency matrix $\mathbf{A}$ and the attention matrix. We perform the softmax operation, and matrix multiply this with $\mathbf{V}$:

$$\mathbf{O} = \text{softmax}(\mathbf{A}^\top \circ \frac{\mathbf{Q}\mathbf{K}^\top}{\sqrt{h_s}}) \cdot \mathbf{V}. \tag{3}$$

Importantly, and in contrast with the CFCN, we do not modify $\mathbf{A}$ to include self-connections / an identity diagonal after the first layer/block (see below). The self-connections are not needed in CaT. Overall, the process presented in Equation 3 is undertaken in parallel according to the number of heads in the Causal Multi-Head Cross-Attention (CMCA), and these outputs are concatenated whilst maintaining the independence between the input variables:

$$\mathbf{O}_{\mathrm{CMCA}} = \mathrm{CMCA}(\boldsymbol{\gamma}, \mathbf{X}_E) =$$
$$\sigma(\mathrm{Dropout}(\mathrm{Proj}(\mathrm{Conc}([\mathbf{O}^1(\boldsymbol{\gamma}, \mathbf{X}_E), \mathbf{O}^2(\boldsymbol{\gamma}, \mathbf{X}_E), \tag{4}$$
$$..., \mathbf{O}^H(\boldsymbol{\gamma}, \mathbf{X}_E)])))),$$

where Conc is a concatenation operation (concatenating along the last dimension corresponding with the dimensionality of each head), $H$ is the number of heads in the CMCA, and Proj is a linear projection layer which projects the output to the same dimensionality as the expected head input.

The CMCA forms part of a causal block ('CBlock'), which includes *two* residual/skip connections:

$$\mathbf{O}_{\mathrm{CMCA}}^{r,'} =$$
$$\mathrm{BN}(\mathrm{CMCA}(\mathbf{O}_{\mathrm{Block}}^{r-1}, \mathbf{X}_E) + \mathbf{O}_{\mathrm{Block}}^{r-1}), \tag{5}$$

$$\mathbf{O}_{\mathrm{Block}}^{r} =$$
$$\mathrm{BN}(FF(\mathbf{O}_{\mathrm{CMCA}}^{r,'}) + \mathbf{O}_{\mathrm{CMCA}}^{r,'}), \tag{6}$$

where BN is (optional) batch normalization (Ioffe & Szegedy, 2015), and $FF$ is a two-layer perceptron with dropout and a Swish (Ramachandran et al., 2018) activation function. At each layer, the embedded input $\mathbf{X}_E$ is always fed to the head(s), whilst the learned embedding $\boldsymbol{\gamma}$ is used only for the head(s) in the first layer, i.e. $\mathbf{O}_{\mathrm{Block}}^r = \boldsymbol{\gamma}$ for $r = 0$. Finally, we stack a number of such blocks in sequence (corresponding with the number of 'layers') and include one last linear layer before the final output.

**Summary of Key Modifications** : The key modifications to the transformer architecture are:

- The application of the cross-attention mask to constrain the network to respect the underlying DGP / DAG.

- The initialization of the learned embedding $\boldsymbol{\gamma}$ which forms the query vector.

- Feeding $\mathbf{X}_E$ at every layer and every block so that the downstream $\boldsymbol{\gamma}$ (which becomes $\mathbf{O}_{\mathrm{Block}}^r$) can attend to both its prior state and the observed embedded input. This allows the model to account for the surprise in the observed value compared to the current estimate. Usually, cross-attention is between two different but informative embeddings. In our case, it is between a source of information $\mathbf{X}_E$ and an empty embedding $\boldsymbol{\gamma}$ which becomes the output as it propagates through the network. Feeding in the input at each layer enables CaT to 'compare' the current $\mathbf{O}_{\mathrm{Block}}^r$ against the input to see what it needs to attend to in the input (whilst constraint according to the DAG) to produce a useful output.

- We do *not* include layer normalization (Ba et al., 2016). Despite its successful integration with transformers, layer normalization is not used as it would rescale potentially calibrated interventions.

| Model | MSE (± Std) | eATE (± Std) |
|---|---|---|
| CaT False DAG 1 | **0.555 (± 0.003)** | 2.314 (± 0.010) |
| CaT False DAG 2 | 0.557 (± 0.004) | 1.382 (± 0.147) |
| CaT True DAG | 1.004 (± 0.007) | **0.058 (± 0.056)** |
| CFCN False DAG 1 | 0.622 (± 0.200) | 2.330 (± 0.056) |
| CFCN False DAG 2 | 0.568 (± 0.008) | 1.335 (± 0.047) |
| CFCN True DAG | 1.023 (± 0.014) | **0.149 (± 0.309)** |
| Rand. Forest | 0.622 (± 0.004) | 2.311 (± 0.012) |
| Transformer | 0.615 (± 0.012) | 2.379 (± 0.057) |
| MLP | 0.559 (± 0.006) | 2.325 (± 0.013) |

Table 1: Simulation results for motivating example highlighting that models which make correct structural assumptions do not necessarily yield predictions with the lowest Mean Squared Error on the test set (often incorrect causal assumptions result in higher predictive capacity, but lower robustness), and that lowest absolute error on the estimation of the Average Treatment Effect (eATE) can only be achieved with the correct causal graph.

### 3.3 Causal Fully-Connected Networks (CFCNs)

The CFCN is constructed similarly to an autoencoder, but only insofar as the inputs are also the targets (see Fig. 5). Targets cannot be predicted from themselves, but only by their structural/causal parents. The output is therefore reflective of the conditional distribution implied by the DAG. For each layer $r$, where $R$ is the total number of layers, we pre-allocate a mask $\mathbf{M}_r$ of zeros of shape $q_r \times p_r$, where $p_r$ is the number of neurons in the layer before layer $r$, and $q_r$ is the number of neurons in the current layer. We assert that the number of neurons in each layer is never less than the dimensionality of the input to avoid bottleneck issues resulting in a dropout of signal along the depth of the network. Furthermore, we assume that the width of each layer is an integer multiple or factor of the preceding layer size, such that when growing or shrinking the network across its depth, the number of neurons capable of interacting with a particular input is always equal across inputs and always greater than or equal to 1. Taking the adjacency matrix we reshape/expand/shrink it to reflect the shape of the underlying weight matrices for each layer, whilst respecting the original dependencies encoded by the associated DAG. The masking process is applied similarly to (Germain et al., 2015):

$$\mathbf{o}_r = \sigma(\mathbf{o}_{r-1}(\mathbf{W}_r \circ \mathbf{M}_r) + \mathbf{b}_r), \tag{7}$$

where $\sigma$ is an activation function, $\mathbf{o}_r$ is the vector output, $\mathbf{W}_r$ is the weight matrix, $\mathbf{b}_r$ is a bias vector, and $\mathbf{M}_r$ is a mask for a given layer $r$, respectively. $\circ$ is the Hadamard product. An example is shown in Sec. S3 and Fig. 4 in the supp. mat. One key element illustrated with two examples in Fig. 5 is the introduction of the identity diagonal to the DAG for all layers greater than 1. As such, the first layer acts as the 'barrier' to prevent inputs from attending to themselves, but once an intermediate (and causally valid) interaction has occurred, any interactions can then be passed on to the output. The identity cannot be introduced into the first layer, because this would enable the network to simply pass through the input to the output, and also violate the fact that variables without parents are exogenous. In contrast, without the introduction of the identity in the second layer, no signals could be propagated from intermediate, causally valid interactions. In Sec. S4 we present CFCN 'mod', which is an alternative we find to perform similarly.

### 3.4 Training and Loss Functions

We use per-variable loss functions, and users are required to specify the variable type as continuous (mean-squared error) or binary (binary cross-entropy). If embeddings are used, the variables are likely to be continuous, but for more specific tasks we provide the option to tailor the loss function to the variable. Importantly, the loss functions can be determined independently of the causal constraint, and other loss functions (e.g. with custom regularizers) can be used as long as the independence constraints implied by the DAG are not violated. For any endogenous node $z$: if $z$ is continuous, we use mean squared error on its output head; if binary, Bernoulli cross-entropy; if categorical, softmax cross-entropy; if a vector embedding, MSE on that vector. The experimenter is thus free to implement the loss they wish, understanding that the predictions are constrained according to the assumed DAG. A strength of CaT is that there are no special losses or regularizers.

### 3.5 Estimands, Interventions, and Recursive Inference

Constraining models with DAGs not only improves robustness to covariate shift but also enables estimation of intervention effects. Consider the mediating graph $D \to M \to Y$ ($D$ is treatment, $Y$ an outcome, and $M$ a mechanism), together with $D \gets L \to Y$ where $L$ (e.g., age) affects both treatment choice and recovery. The latter paths confound the estimation of $D$'s effect on $Y$. Using $do$-calculus (Pearl, 2009), the interventional distribution $Y \mid do(D = 1)$ can be identified as $Y \mid D = 1, L$, since conditioning on $L$ removes the spurious $L \to D$ path. Importantly, $M$ must be excluded: conditioning on $M$ would make $D$ redundant because $M$ is a more proximal cause. This is a case of transitive reduction (Richardson et al., 2017; Verma & Pearl, 1990; Vowels, 2023b), where intermediate nodes can be dropped if they do not affect relevant conditional independencies. Thus the chain $D \to M \to Y$ simplifies to $D \to Y$ for estimating the total effect.

CaT supports all conditional average (CATE) and averagage (ATE) causal effect queries identifiable from the DAG via standard identification, operationalized as follows: set intervened variables to their intervention values, then recursively update descendants in topological order by re-evaluating the corresponding node heads with parents fixed to their current values, following the g-formula

(Robins, 1986). Using the DAG, this algorithm iteratively updates downstream children of the intervention variables to propagate effects. Full details are given in Sec. S6 and Alg. 1 in supp. mat.

## 4 EXPERIMENTS

We provide results for five experiments: (a) a simulated dataset motivating the need for CaTs, (b) causal inference benchmark datasets Jobs (LaLonde, 1986; Smith & Todd, 2005), Twins (Almond et al., 2005) and (c) a real-world application of CaT to psychology data during the COVID-19 pandemic (McBride et al., 2021; Vowels et al., 2023a) highlighting its flexibility in leveraging multi-dimensional inputs. Note that owing to space constraints some of these details concerning these evaluations are presented in Sections S10 and S11 in supp. mat. No hyperparameter tuning was undertaken for these experiments - again, the goal is not to provide a model which excels at a particular task, but to provide a general modeling class which facilitates the integration of structural constraints across a range of tasks. We include CFCN as a baseline as it isolates the value of structural masking but *without attention*. Using the same DAG mask with layered identity handling, it serves as an architectural control to show that benefits are not unique to the attention mechanism of the transformer.

| | Twins | | Jobs | | | |
| Model | eATE ws | eATE os | Risk ws | Risk os | eATT ws | eATT os |
|---|---|---|---|---|---|---|
| CaT | .0110 ± .0061 | .0133 ± .0078 | .25 ± .02 | .27 ± .06 | .016 ± .01 | .086 ± .06 |
| CFCN | .0098 ± .0048 | .0102 ± .0092 | .25 ± .02 | .26 ± .06 | .016 ± .01 | .084 ± .06 |
| GANITE (Yoon et al., 2018) | .0058 ± .0017 | .0089 ± .0075 | .13 ± .01 | .14 ± .00 | .01 ± .01 | .06 ± .03 |
| CFRwass (Shalit et al., 2017) | .0112 ± .0016 | .0284 ± .0032 | .17 ± .00 | .21 ± .00 | .04 ± .01 | .09 ± .03 |
| BART (Chipman et al., 2010) | .1206 ± .0236 | .1265 ± .0234 | .23 ± .00 | .25 ± .02 | .02 ± .00 | .08 ± .03 |
| C Forest (Wager & Athey, 2018) | .0286 ± .0035 | .0335 ± .0083 | .19 ± .00 | .20 ± .02 | .03 ± .01 | .07 ± .03 |
| CEVAE (Louizos et al., 2017a) | - | - | .15 ± .00 | .26 ± .00 | .02 ± .00 | .03 ± .01 |
| TVAE (Vowels et al., 2021b) | - | - | .16 ± .00 | .16 ± .00 | .01 ± .00 | .01 ± .00 |

Table 2: Results ± standard deviations for the Twins and Jobs benchmark datasets. eATE - absolute error on the estimation of the Average Treatment Effect; Risk - Policy risk; eATT - absolute error on the estimation of the Average Treatment effect on the Treated; ws - within sample; os - out of sample.

**Toy Motivating Example:** We illustrate the advantage of causal constraints with a simple example adapted from (Hilpert & Vowels, 2024). Data are generated according to the graph in Fig. 3a. The task is to estimate the *total effect* of $D$ on $Y$, and to evaluate both the absolute error on this estimate (eATE) and the Mean Squared Error (MSE) of predicting $Y$. We compare three conditions: the true graph (Fig. 3a), a graph reflecting typical machine learning assumptions (Fig. 3b), and a graph with a missing and a reversed edge (Fig. 3c). CFCN and CaT are evaluated alongside a standard transformer (Vaswani et al., 2017; Paszke et al., 2017), random forest (Breiman, 2001; Pedregosa et al., 2011), and multilayer perceptron. Results are shown in Table 1 and Fig. 1. No hyperparameter tuning was applied to any model.

Three observations follow. First, CaT and CFCN achieve the lowest eATE under the true graph, as expected when respecting the DGP. Second, with the correct causal graph, models also attain the lowest MSE, though in general non-causal predictors can sometimes yield better accuracy than a faithful causal structure. Notably, CaT and CFCN perform comparably to other methods when the graph is misspecified. Third, Fig. 1 shows that non-causal models, while potentially achieving lower MSE, are highly sensitive to covariate shift. In contrast, CaT and CFCN remain stable, reflecting robustness that extends beyond causal inference tasks.

**Causal Inference Tasks**: Results for the three benchmark datasets are shown in Table 2. As noted by recent authors (Vowels et al., 2023b; Curth et al., 2021), such benchmarks have limitations, so our interpretations remain cautious (see also Sec. S12). Evaluation metrics are detailed in Sec. S11. Lower scores indicate better performance. Since most comparison methods are designed and tuned specifically for causal inference, we do not expect CaT or CFCN to excel. Nevertheless, both achieve performance comparable to specialized methods. This result is notable given our minimal use of inductive bias. For the Twins and Jobs datasets, the only assumption is that all non-treatment/non-outcome variables may be confounders. *This graph is implicitly assumed by other methods*, which treat all covariates equally. In contrast, TVAE, CFRWass, DML, and TL incorporate additional

(a) True DAG   (b) False DAG 1  (c) False DAG 2

Figure 3: Motivating example. The data for the experiment are generated from DAG (a), whilst (b) and (c) represent to misspecified DAGs used for estimation. Dashed curved lines in (b) emphasise possible endogeneity of $D$, $L_1$ and $L_2$, and the absence of independence in typical non-causal machine learning approaches. The thick edges depict the target causal relationship of interest.

domain-specific techniques, giving them a natural advantage. By avoiding such tailoring, our approach is at an inherent disadvantage, yet CaT and CFCN remain competitive, underscoring their potential.

**Real-World Psychology:** We also conducted a secondary analysis of UK data from wave 2 of the COVID-19 Psychological Research Consortium Study (McBride et al., 2021), collected during April/May 2020. After excluding cases with missing attachment data, 895 participants remained. The original study examined effects of attachment style on compliance and mental health outcomes. Following (Vowels et al., 2023a), we estimated the causal effect of shifting attachment styles on depression, but using the full multidimensional set of questionnaire items. We compared a naive bivariate linear model, a targeted learning (TL) estimator with semi-parametric techniques, and CaT. The DAG from (Vowels et al., 2023a), derived from causal discovery and domain expertise, was applied (see Sec. S11). Competing methods had to operate on aggregated scale scores, whereas CaT used all questionnaire items. Using secure attachment as the reference, we estimated effects of fearful, anxious, and avoidant styles on depression. TL and the naive model yielded standard errors directly, while CaT's were obtained via 100 bootstrap iterations. Despite not being optimized for causal inference, CaT produced results broadly consistent with TL, without hyperparameter search. As with any real-world causal analysis, however, no ground truth exists, so these findings serve primarily as a demonstration of applicability.

## 5   LIMITATIONS & DISCUSSION

One principal limitation of CaTs concerns the recursive inference procedure. For DAGs which include long mediation structures, and when the intervention node(s) is(are) early in the causal ordering, the recursion can lead to compounding error as the prediction for each mediator is fed back into the model for the prediction of the next descendant. This issue can be alleviated with transitive reduction (Richardson et al., 2017; Verma & Pearl, 1990; Vowels, 2023b) to remove nodes which are not principally important for inference or downstream tasks. Additionally, in many application areas, it may not be possible to specify a DAG because either the phenomenon is too complex, or because the DGP us unknown. However, in such cases, users can either leverage causal discovery techniques (Vowels et al., 2022; Heinze-Deml et al., 2018; Glymour et al., 2019; Spirtes & Zhang, 2016) and work with a putative graph, and otherwise err on the side of caution by including more rather than fewer edges in the DAG (the removal of an edge represents a stronger assumption than an inclusion). Furthermore, CaTs are flexible in that they can be used both for causal inference tasks requiring confident knowledge of the DGP, but equally for prediction, where the incorporation of inductive biases can be beneficial in building robust models.

CaT has yet to be scaled to and tested on more complex computer vision tasks with pre-defined DAGs (such as autonomous driving or human-to-human interactions). Regardless, CaT is agnostic to the integration of various scaling techniques for efficient attention (incl. Flash Attention (Dao et al., 2022; Dao, 2024), Hedgehog & the Porcupine (Zhang et al., 2024), or Performer (Choromanski et al., 2020)).

| Group | Naive | TL | CaT |
|-------|-------|-----|-----|
| Fearful | $.13 \pm .02$ | $.05 \pm .01$ | $.01 \pm .02$ |
| Anxious | $.12 \pm .02$ | $.05 \pm .02$ | $.02 \pm .03$ |
| Avoidant | $.01 \pm .01$ | $.01 \pm .01$ | $.03 \pm .04$ |

Table 3: Bootstrapped effect sizes $\pm$ standard errors for impact of change of attachment style on levels of depression. Compares naive (bivariate) correlation, Targeted Learning (TL), and CaT. The naive and TL based approaches use averages of the construct/scale/questionnaire items, whereas CaT uses the full questionnaire without aggregation of questionnaire items.

## 6 CONCLUSION

Transformers are powerful function approximators but lack robustness to covariate shift and cannot naturally encode structural inductive biases, limiting their reliability in scientific applications where robustness and interpretability are critical. We addressed this by introducing the Causal Transformer (CaT), which integrates domain knowledge via DAG-based constraints, and a comparable baseline (CFCN). These models combine the flexibility of neural networks with explicit causal structure, while remaining agnostic to efficiency improvements such as Flash attention. Importantly, CaT and CFCN are complementary to methods like Double Machine Learning (DML; (Chernozhukov et al., 2018; 2017)) and Inverse Probability Weighting (IPW; (Rosenbaum & Rubin, 1983)), and can serve as plug-in estimators. This makes them a versatile addition to the empirical researcher's toolbox.

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

$$
\begin{array}{c|ccc}
 & X^1 & X^2 & X^3 \\
\hline
X^1 & 0. & 1. & 1. \\
X^2 & 0. & 0. & 1. \\
X^3 & 0. & 0. & 0.
\end{array}
\rightarrow
\begin{bmatrix}
0. & 1. & 1. \\
0. & 0. & 1. \\
0. & 0. & 0.
\end{bmatrix}^{3\times3}
\rightarrow
\begin{bmatrix}
1. & 1. & 1. & 1. & 1. & 1. \\
0. & 0. & 1. & 1. & 1. & 1. \\
0. & 0. & 0. & 0. & 1. & 1.
\end{bmatrix}^{3\times6}
\rightarrow
\begin{bmatrix}
1. & 1. & 1. \\
1. & 1. & 1. \\
0. & 1. & 1. \\
0. & 1. & 1. \\
0. & 0. & 1. \\
0. & 0. & 1.
\end{bmatrix}^{6\times3}
$$

Adj. Matrix   Layer 1 Mask (neurons=3)   Layer 2 Mask (neurons=6)   Layer 3 Mask (neurons=3)

Figure 4: Simple example sequence of matrix transformations used to generate masks for CFCN in a three variable, three layer case, where the DAG includes $X^1 \rightarrow X^2 \rightarrow X^3$ mediation as well as a direct path $X^1 \rightarrow X^3$, and the number of neurons in each layer is 3, 6, and 3. The transition from Layer 1 to Layer 2 includes the introduction of the diagonal 'pass-through' which is absent in the first layer.

## SUPPLEMENTARY MATERIAL

## S1. OVERVIEW

This supplementary material accompanies the paper CaTs and DAGs and full code, including experiments, is attached. An open repository containing the code will also be released upon acceptance. We begin this supplementary material by providing a list of notation used throughout, in Section S2. We then provide some additional illustrations and information concerning the masking mechanisms used in CFCN in Section S3 and describe a variation of CFCN called CFCN 'mod' in Section S4. We provide an overview of the typical self- and cross-attention mechanisms used in transformers in Section S5, and present the algorithm used for recursive inference in CaTs and CFCNs in Section S6. We describe an approach called Isomorphic Shuffling in Section S7, although this was not found to yield good results in practice.

In Section S8 we present some preliminaries relating to causal inference including the notion of one-step-ahead potential outcomes (which motivates the recursive inference algorithm in Section S6). Then, in Section S9 we provide the hyperparameters used for the evaluations. In Section S10 we provide additional information and results for the motivating simulations which were presented in the main text. In Section S11 we provide additional experimental results for other causal inference tasks involving the benchmark Jobs and Twins benchmark datasets as well as the real-world psychology dataset.

In Section S12 we present a curious result for how the same scikit-learn (Pedregosa et al., 2011) random forest (Breiman, 2001) and transformer used in the motivating experiments perform on the causal inference benchmarks.

Finally, in Section S13 we provide a theoretical proof of why the architectural design ensures that the conditional independencies implied by the DAG hold in the output of the network.

## S2. NOTATION

- $\mathcal{D}$ = dataset
- $N$ = sample size
- $z$ = node in set of nodes in graph
- $Y$ = univariate outcome
- $\mathbf{Y}$ = multivariate outcome
- $L$ = univariate covariate
- $\mathbf{L}$ = multivariate covariate
- $\mathbf{L}_{\text{set}}$ = set of multivariate covariates
- $|\mathbf{L}|$ = number of covariates

- $D$ = univariate treatment
- $\mathcal{P}$ = joint observational distribution
- $\mathcal{P}(Y|D, L)$ = conditional observational distribution
- $\boldsymbol{\tau}$ = treatment effect
- $\mathcal{G}$ = topologically sorted DAG
- $\mathbf{A}$ = adjacency matrix for DAG
- $Z$ = set of nodes in DAG
- $|Z|$ = number of nodes in DAG
- $\mathcal{O}$ = causal ordering for each variable in topologically sorted DAG
- $\mathcal{F}$ = causal descendants of intervention nodes
- $\mathcal{D}'$ = dataset updated with interventions and predictions of estimated effects
- CFCN = Causal Fully Connected Network
- $r$ = layer or block number in CFCN or CaT, respectively
- $R$ = total number of layers or blocks in CFCN or CaT, respectively
- $\mathbf{W}_r$ = weight/parameter matrix in layer $r$
- $\sigma$ = activation function
- $\mathbf{b}_r$ = bias vector for layer $r$
- $\mathbf{o}_r$ = vector output of CFCN for layer $r$
- $\mathbf{M}_r$ = CFCN parameter mask for layer $r$
- CaT = Causal Transformer
- $\mathbf{Q}$ = query embeddings in CaT and conventional transformer
- $\mathbf{K}$ = key embeddings in CaT and conventional transformer
- $\mathbf{V}$ = value embeddings in CaT and conventional transformer
- $\mathbf{S}$ = 'similarities' in conventional transformer
- $\mathbf{X}$ = input to CaT
- $\mathbf{X}_E$ = 're-embedded' input $\mathbf{X}$ in CaT
- $C$ = embedding dimension for CaT input before 're-embedding'
- $d_E$ = dimensionality after 're-embedding'
- $\boldsymbol{\gamma}$ = learnable embedding
- $H$ = number of parallel heads in each block
- $h_s$ = head size
- CMCA = Causal Multi-head Cross Attention
- $\mathbf{O}_{\mathrm{CMCA}}$ = output of CMCA
- $\mathbf{O}^1, ... \mathbf{O}^H$ = output of parallel heads
- $\mathbf{O}^r_{\mathrm{Block}}$ = output from CaT layer/block $r$
- $P$ = permutation matrix
- $\mathcal{I}$ = set of all intervention value and variable pairs $(\mu, z)$
- $(\mu, z)$ = an intervention value and variable pair, respectively
- $\mathcal{M}$ = Trained CaT or CFCN model
- $\hat{Q}$ = empirical estimator (e.g. random forest or CaT)

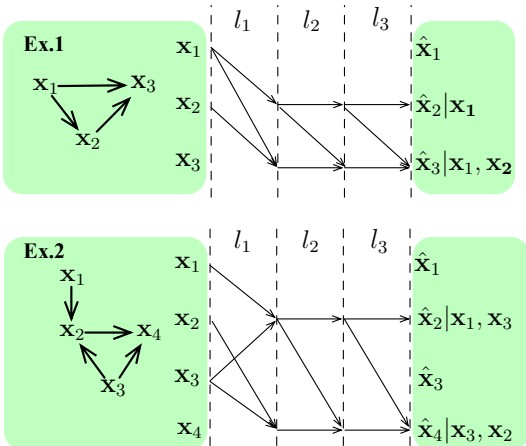

Figure 5: Two examples demonstrating how the delayed introduction of the identity diagonal at layer 2 and onwards (a) prevents inputs attending to themselves at the first layer and (b) allows other intermediate predictions of these inputs to be used for the prediction other variables. Without the introduction of identity, signals from e.g. $\mathbf{x}_1$ used for predicting $\mathbf{x}_2$ would be blocked, and $\mathbf{x}_2$ itself could not be predicted. Applies to both CaT and CFCN.

## S3. CFCN MASKING

The masking process in CFCN is applied similarly to (Germain et al., 2015):

$$\mathbf{o}_r = \sigma(\mathbf{o}_{r-1}(\mathbf{W}_r \circ \mathbf{M}_r) + \mathbf{b}_r), \tag{8}$$

where $\sigma$ is an activation function, $\mathbf{o}_r$ is the vector output, $\mathbf{W}_r$ is the weight matrix, $\mathbf{b}_r$ is a bias vector, and $\mathbf{M}_r$ is a mask for a given layer $r$, respectively. $\circ$ is the Hadamard product. An example is shown in Figure 4. A visualization of this process across different layers is provided in Fig.5. Note that, while not strictly necessary for respecting the DAG, links between paths (such as the diagonal links visible in Fig.5) improve network expressivity by enabling variables like $x_1$ and $x_2$ to interact more deeply across layers, e.g. recursively processing $(x_1, x_2)$ together to influence $x_3$.

|  | **Twins** |  | **Jobs** |  |  |  |
| Model | eATE ws | eATE os | Risk ws | Risk os | eATT ws | eATT os |
|---|---|---|---|---|---|---|
| CFCN | .0098 ± .0048 | .0102 ± .0092 | .25 ± .02 | .26 ± .06 | .016 ± .01 | .084 ± .06 |
| CFCN mod | .0049 ± .0050 | .0065 ± .0066 | .24 ± .02 | .27 ± .06 | .011 ± .01 | .082 ± .06 |

Table 4: Results ± standard deviations for the CFCN 'mod' approach described in Sec. S4 for the Twins and Jobs benchmark datasets. eATE - absolute error on the estimation of the Average Treatment Effect; Risk - Policy risk; eATT - absolute error on the estimation of the Average Treatment effect on the Treated; ws - within sample; os - out of sample.

## S4. CFCN MOD

The CFCN Mod is a variation to the previously introduced CFCN described in Sec. 3.3, wherein each layer has direct access to the input through a linear connection. The computation of the layers is given by

$$\begin{aligned} \mathbf{y}_0 &= \sigma(\mathbf{x}(\mathbf{W}_0 \circ \mathbf{M}_0) + \mathbf{b}_0), \\ \mathbf{y}_r &= \sigma(\mathbf{y}_{r-1}(\mathbf{W}_r^* \circ \mathbf{M}_r^*) + \mathbf{x}(\mathbf{W}_r \circ \mathbf{M}_r) + \mathbf{b}_r), \end{aligned} \tag{9}$$

where $x$ is the input vector, $\mathbf{y}_r$ is an output vector, $\mathbf{W}_r, \mathbf{W}_r^*$ are weight matrices for layer $r$, $\mathbf{M}_r$, $\mathbf{M}_r^*$ are the corresponding masks, $\mathbf{b}_r$ is a bias vector, and $\circ$ is the Hadamard product.

A key distinction from the CFCN architecture lies in the handling of masks for the weight matrices. Specifically, the masks $\mathbf{M}_r^*$ include the identity diagonal, ensuring that each layer maintains

self-connections. Conversely, the mask for $\mathbf{M}_r$ does not include this diagonal, preventing direct propagation of the input to the output, and enforcing the DaG constraints.

These modifications ensure that each intermediate layer has access to the input, thus mitigating the risk of input signal degradation throughout the network. Additionally, this design facilitates a meaningful comparison between the intermediate output features and the original input. For instance, if we consider a model predicting weight from height, the difference between the predicted weight at an intermediate layer and the true weight (derived from the input) could serve as a valuable indicator of body composition, such as identifying obesity. We hypothesize that this structure enhances the model's ability to retain essential input information while allowing richer feature interactions across layers.

Table 4 show the performance of the CFCN and CFCN mod on the Twins and Jobs benchmark datasets. Whilst we believe CFCN 'mod' is a valuable alternative to CFCN, our current evaluations indicate similar performance across the two variations.

## S5. CONVENTIONAL MULTI-HEAD SELF- AND CROSS-ATTENTION

The core components of transformers (Vaswani et al., 2017) concern the way inputs are processed by the multi-head attention (MHA). Consider an input of $|Z| = 3$, $C$-dimensional embeddings $\mathbf{a}$, $\mathbf{b}$, and $\mathbf{c}$ (boldfont to indicate vectors). Together these variables constitute an input matrix $\mathbf{X}$, which is $B \times |Z| \times C$. Each of the three constituent inputs in $\mathbf{X}$ are processed independently, and by three linear transformations, yielding the 'query' $\mathbf{Q}$, 'key' $\mathbf{K}$, and 'value' $\mathbf{V}$ embeddings. Self-attention, derived via $\mathbf{S} = \left( \frac{\mathbf{Q}\mathbf{K}^\top}{h_s^{0.5}} \right)$ where $h_s$ is the dimensionality of the embeddings, creates a 3x3 interaction matrix, describing the extent to which each pair from the embeddings of $\mathbf{a}$, $\mathbf{b}$ and $\mathbf{c}$ are similar to each other. This attention matrix represents the key opportunity for causal masking with an adjacency matrix, and we discuss how this is done below. The attention matrix is then matrix multiplied with $\mathbf{V}$ such that the different components of $\mathbf{V}$ interact with a strength proportional to the similarities encoded in $\mathbf{S}$.

In addition to self-attention, transformers may also employ cross-attention mechanisms, which can be useful in contexts where the input and targets differ. Cross-attention processes two distinct sets of embeddings: a query matrix $\mathbf{Q}$ from the target sequence and key-value matrices $\mathbf{K}$ and $\mathbf{V}$ from the input sequence. Suppose we have a target matrix $\mathbf{T}$ of size $B \times |Z'| \times C$, where $|Z'|$ is the number of target embeddings. The query $\mathbf{Q}$ is obtained from $\mathbf{T}$, while the key $\mathbf{K}$ and value $\mathbf{V}$ are derived from the input matrix $\mathbf{X}$. Cross-attention then computes the similarity between the target queries and the input keys using $\mathbf{S} = \left( \frac{\mathbf{Q}\mathbf{K}^\top}{h_s^{0.5}} \right)$, resulting in a $|Z'| \times |Z|$ interaction matrix. This interaction matrix represents how much each target embedding in $\mathbf{T}$ attends to the input embeddings in $\mathbf{X}$. As with self-attention, $\mathbf{S}$ is then used to reweight $\mathbf{V}$, allowing information from the input to be incorporated into the target sequence based on the learned interactions.

## S6. INFERENCE ALGORITHM

The algorithm for performing recursive inference with CaT or CFCN is shown in Alg. 1. This algorithm can be read as an implementation of *recursive substitution* (see Section S8.)

## S7. ISOMORPHIC SHUFFLING

We develop a means to shuffle the ordering of the variables and the masks such that the topology of causal structure remains the same whilst the order of the adjacency matrix is permuted randomly during training. This approach is inspired by the order agnostic training presented by (Germain et al., 2015). These permutations are isomorphic with respect to the DAG used to specify the adjacency matrix, and to the weights/parameters of the CFCN and CaT which are not shuffled. The function approximation process is thereby forced to yield a solution which respects the constraints of the underlying DAG, whilst nonetheless being forced to fulfil this role under permutations of this DAG.

---

**Algorithm 1** Estimating the effects of interventions.

---
1: **function** FORWARD($\mathcal{D}, \mathcal{I}, \mathcal{M}, \mathcal{A}, \mathcal{O}$)
2:     **input:** $\mathcal{D}$ - dataset
3:     **input:** $\mathcal{I}$ - intervention value and variable pairs $(\mu, z)$
4:     **input:** $\mathcal{M}$ - Trained CaT or CFCN model
5:     **input:** $A$ - topologically sorted adjacency matrix for DAG
6:     **input:** $\mathcal{O}$ - causal ordering for each variable in DAG
7:     **output:** $\mathcal{D}'$ - interventions and estimated effects
8:     $\mathcal{D}' \leftarrow \text{copy}(\mathcal{D})$         $\triangleright$ Create a copy of the data
9:     $I \leftarrow \{\mu : (\mu, z) \in \mathcal{I}\}$         $\triangleright$ Intervention nodes
10:     **for** $\mu, z$ in $\mathcal{I}$ **do**
11:         $\mathcal{D}'(\mu) \leftarrow z$         $\triangleright$ Apply intervention
12:     **end for**
13:     $\mathcal{F} \leftarrow \text{Desc}(I, A) \setminus I$         $\triangleright$ Get nontrivial causal descendants
14:     $\mathcal{F} \leftarrow \text{Sort}(\mathcal{F}, \mathcal{O})$         $\triangleright$ Topological sort
15:     **for** $\mu_d$ in $\mathcal{F}$ **do**         $\triangleright$ Iterate through sorted descendants
16:         $\text{preds} \leftarrow \mathcal{M}(\mathcal{D}')$         $\triangleright$ Get model predictions
17:         $\mathcal{D}'(\mu_d) \leftarrow \text{preds}$         $\triangleright$ Recursively update dataset.
18:     **end for**
19:     **return** $\mathcal{D}'$
20: **end function**

---

More formally, assume that $A$ is the adjacency matrix for the DAG representing the DGP. $A$ is $|Z| \times |Z|$ if the DAG has $|Z|$ nodes, where $A_{ij} = 1$ if there is an edge between vertex $i$ and vertex $j$. We generate a random ordering $p$ from 1 to $|Z|$, and use this to construct a permutation matrix $\mathbf{P}$, which is also $|Z| \times |Z|$. The $i$th row of $P$ is assigned a 1 in the $j$th column if, according to the random ordering $p$, vertex $i$ should be mapped to vertex $j$ in the permutation. The permuted (but isomorphic to the original graph) adjacency matrix $\mathbf{A}' = \mathbf{P}\mathbf{A}\mathbf{P}^\top$.

Note that in practice, we find that this shuffling worsened both predictive metrics (like MSE) and widened the standard errors for the ATE estimates. However, it found positive application in (Germain et al., 2015) and (Uria et al., 2014), so it may be found to be useful for specific tasks.

## S8. CAUSAL PRELIMINARIES

We consider the set of variables $\mathbf{Z}_{\text{set}}$, where $\mathbf{Y}, \mathbf{D} \subset \mathbf{Z}_{\text{set}}$ denote the (possibly multi-dimensional and/or multi-variate) outcome and treatment, respectively, and $\mathbf{L}_{\text{set}} = \mathbf{Z}_{\text{set}} \setminus \mathbf{Y}, \mathbf{D}$ are covariates. We assume a causal ordering $\mathcal{O}$ among these variables, and that the causal relations between them are represented by a DAG $\mathcal{G}$. For any $\mathbf{Z} \subseteq \mathbf{Z}_{\text{set}}$, we denote by $\text{Pa}_{\mathbf{Z}}$ the set of vertices (corresponding to variables) that have a directed edge to $\mathbf{Z}$ in $\mathcal{G}$ (direct causes, or parents). $\mathcal{G}$ conforms to the ordering $\mathcal{O}$ in the sense that $\text{Pa}_z$ precede $z$ in the ordering $\mathcal{O}$ for any $z \in \mathbf{Z}_{\text{set}}$ – this indeed guarantees that $\mathcal{G}$ is a DAG and no cycles are formed.

For ease of notation, we will use the same notation for a graph vertex as for its value. Following (Malinsky et al., 2019; Shpitser et al., 2022), we let the one-step-ahead *potential outcome* variables $\mathbf{Z}(\text{Pa}_{\mathbf{Z}})$ denote the value of $\mathbf{Z}$ given its parents values $\text{Pa}_{\mathbf{Z}}$, possibly contrary to fact. The latter can also be defined as the output of a structural equation model (SEM) where a function $f(\cdot)$ maps the values of $\text{Pa}_{\mathbf{Z}}$ along with an exogenous noise $\epsilon$ to values of $\mathbf{Z}$. We assume the existence of all one-step-ahead potential outcomes. Given these one-step-ahead potential outcomes, for any subset $\mathbf{D} \subset \mathbf{Z}_{\text{set}}$, the potential outcome $\mathbf{Z}(\mathbf{D})$, is defined through *recursive substitution* as follows:

$$\mathbf{Z}(\mathbf{D}) := \mathbf{Z}\Big(\text{Pa}_{\mathbf{Z}} \cap \mathbf{D}, \;\; \text{Pa}_{\mathbf{Z}} \setminus \mathbf{D}\,(\mathbf{D})\Big). \tag{10}$$

When predicting the values of $\mathbf{Z}(\mathbf{D})$ in Alg. 1, we apply recursive substitution by first predicting the values of $\text{Pa}_{\mathbf{Z}} \setminus \mathbf{D}\,(\mathbf{D})$ and plugging them into Eq. equation 10 for each variable $\mathbf{Z}$.

Equivalent to what we described above, $\mathbf{Z}(\mathbf{D})$ can be defined as the random variable induced by a modified SEM where the functions $f(\cdot)$ for those variables in $\mathbf{D}$ are replaced by constant functions setting their value to $\mathbf{d}$. Authors denote this modified SEM using $do(\mathbf{D} = \mathbf{d})$ following Pearl's *do* notation (Pearl, 2009). In particular, the distribution of $\mathbf{Z}(\mathbf{D})$ is shown by $\mathcal{P}(\mathbf{Z}|do(\mathbf{D} = \mathbf{d}))$.

For outcome $\mathbf{Y}$ and treatment $\mathbf{D}$, the average treatment effect is defined as:

$$\boldsymbol{\tau} := \mathbb{E}_{\mathbf{Y} \sim P(\mathbf{Y}|do(\mathbf{D}=\mathbf{1}))}[\mathbf{Y}] - \mathbb{E}_{\mathbf{Y} \sim P(\mathbf{Y}|do(\mathbf{D}=\mathbf{0}))}[\mathbf{Y}].$$

For any subset $\mathbf{L}_{\text{sub}} \subseteq \mathbf{L}_{\text{set}}$ such that $\mathbf{Y}(\mathbf{d})$ is independent of $\mathbf{D}$ conditioned on $\mathbf{L}_{\text{sub}}$ (equivalently an adjustment set in Pearl's terminology)[2], we have

$$
\begin{aligned}
\boldsymbol{\tau} &= \mathbb{E}_{\mathbf{L}_{\text{sub}} \sim \mathcal{P}} \Big[ \mathbb{E}_{\mathbf{Y} \sim P(\mathbf{Y}|do(\mathbf{D}=\mathbf{1}),\mathbf{L}_{\text{sub}})}[\mathbf{Y}] \\
&\qquad - \mathbb{E}_{\mathbf{Y} \sim P(\mathbf{Y}|do(\mathbf{D}=\mathbf{0}),\mathbf{L}_{\text{sub}})}[\mathbf{Y}] \Big] \\
&= \mathbb{E}_{\mathbf{L}_{\text{sub}} \sim \mathcal{P}} \Big[ \mathbb{E}_{\mathbf{Y} \sim P(\mathbf{Y}|\mathbf{D}=\mathbf{1},\mathbf{L}_{\text{sub}})}[\mathbf{Y}] \\
&\qquad - \mathbb{E}_{\mathbf{Y} \sim P(\mathbf{Y}|\mathbf{D}=\mathbf{0},\mathbf{L}_{\text{sub}})}[\mathbf{Y}] \Big].
\end{aligned}
\tag{11}
$$

In practice, we might estimate this quantity using an empirical estimator (like a random forest) $\hat{Q}$ as follows:

$$
\begin{aligned}
\hat{\tau}(\hat{Q}; \mathbf{L}_{\text{sub}}) = \frac{1}{N} \sum_{i=1}^{n} \Big[ &\hat{Q}(\mathbf{D}=\mathbf{1}, \mathbf{L}_{\text{sub}} = \mathbf{L}_{\text{sub},i}) \\
&- \hat{Q}(\mathbf{D}=\mathbf{0}, \mathbf{L}_{\text{sub}} = \mathbf{L}_{\text{sub},i}) \Big],
\end{aligned}
\tag{12}
$$

where the hatˆnotation indicates the value is estimated empirically rather than population/true quantity.

Finally, and importantly, we note that the success of causal inference rests on a number of strong and often untestable assumptions (Yao et al., 2020; Guo et al., 2020; Rubin, 2005; Imbens & Rubin, 2015; Grosz et al., 2020; Petersen et al., 2012; Rubin, 1980; Hernan, 2018): (1) Ignorability/Unconfoundedness/Conditional Exchangeability - this assumes that there are no unobserved confounders, meaning that individuals with identical covariates should have the same probability of receiving treatment, and their potential outcomes should also be identical given the same latent covariates. (2) Positivity - this requires that treatment assignment probabilities are strictly positive and not deterministic, ensuring that $P(\mathbf{D} = \mathbf{d}_i | \mathbf{L} = \mathbf{L}_i) > 0, : \forall; \mathbf{d}, \mathbf{L}$. In other words, for every possible subset of the data, there must be a non-zero probability of receiving any given treatment. (3) Stable Unit Treatment Value Assumption (SUTVA) (Rubin, 1980; Laffers & Mellace, 2020; Schwartz et al., 2012) - this assumption states that an individual's potential outcomes are not influenced by the treatment assignments of others, meaning there are no spillover effects, and that the effect of a treatment remains consistent regardless of how an individual came to receive it (Schwartz et al., 2012).

## S9. EXPERIMENTAL SETUP AND HYPERPARAMETERS

The hyperparameters for CaT and CFCN (and, identically, CFCN mod) were the same for the motivating experiment, Twins, Jobs, and the real-world dataset experiments presented in the main text, and are shown in Table 5.

## S10. MOTIVATING EXAMPLE

We generate data from three Data Generating Processes (DGPs), the Directed Acyclic Graphs (DAGs) for which are shown in Figure 3 in the main text. The structural equations for the DGPs for all three graphs are the same (the data are generated according to the correct graph). In contrast, no graphs are given to the transformer, random forest (RF), or multilayer perceptron (MLP). These do

---

[2]Note that such a subset always exists, e.g., $\mathbf{L}_{\text{sub}} = \text{Pa}_{\mathbf{D}}$.

| Method | Hyperparameter | Value |
|--------|----------------|-------|
| **CFCN** | Dropout | 0.0 |
|  | Neurons per layer | [s, 2*s, 2*s, 2*s, s] |
|  | Activation | Swish |
| **CaT** | Dropout | 0.0 |
|  | Num Heads | 2 |
|  | Num Layers | 2 |
|  | Head Size | 6 |
|  | Input Embedding Dim | 5 |
|  | Projection Dim | 6 |
| **Training** | No. Iters CaT | 20k |
|  | No. Iters CFCN | 6k |
|  | Batch Size | 100 |
|  | Learning Rate | 5e-3 |
|  | Optimizer | AdamW (Loshchilov & Hutter, 2019) |

Table 5: Hyperparameters for CFCN, CaT, and Training Settings. Note that 's' is the input size for CFCN which is determined by the dimensionality of the input for the dataset being used.

not accept graph constraints, whereas CFCN and CaT are evaluated under the three different sets of constraints. The structural equations are as follows:

$$
\begin{aligned}
U_D &\sim \mathcal{N}(0,1), \ U_{l_1} \sim \mathcal{N}(0,1), \\
U_{l_2} &\sim \mathcal{N}(0,1), \ U_Y \sim \mathcal{N}(0,1), \\
D &= U_D, \ L_1 = 0.8D + U_{l_1}, \\
Y = 0.8D + 0.4L_1 + U_y, \ L_2 &= 0.7Y + 0.6D + U_{l_2},
\end{aligned}
\tag{13}
$$

and as such, the total average treatment effect (ATE) of $D$ on $Y$ is $(0.8 + (0.8 * 0.4) = 1.12$.

The RF and MLP are implemented in scikit-learn (Pedregosa et al., 2011), where as the transformer utilizes the TransformerEncoder module from pytorch (Paszke et al., 2017), and is given the same hyperparameters as CaT. Given the simplicity of the DGP (4 unidimensional variables in total, linear functional form, Gaussian error) we use the default hyperparameters in the MLP and RF. Indeed, the default parameters in the scikit-learn RF have been shown to work well across a wide range of tasks (Probst et al., 2018). The hyperparameters for the transformer and the CaT are as follows: batch size 300, training iterations 20k, number of heads 3, number of layers 3, head size 4, input embedding dimension 4, 'internal' embedding dimension 4, and learning rate of 0.0002. For CFCN, the batch size was 50, we used 4, 8, and 4 neurons in each of the hidden layers, and applied a learning rate of 0.001. Both CFCN and CaT were trained with an AdamW optimizer (Loshchilov & Hutter, 2019) with learning rate scheduling.

The sample size is set to 10,000 to avoid issues with random sampling variation, and generate data and undertake the analysis for 30 simulations of these data. The MSEs presented in the main text are computed over the validation set which represented a 20% split of the data, and the ATEs were computed over the full training-validation combination, once the models had been trained on thet training data. This paradigm (training on the training set but evaluating the ATE across the full dataset) is normal in the context of causal inference, where the optimization of the network during training does not represent the same task as in conventional supervised learning. In any case, very similar results are obtained if only the validation split is used for evaluating the ATE estimation.

The results for the error on the ATE estimation across all methods are shown in Fig 6. This plot highlights how the lowest error on the estimation of the ATE is provided by CaT and CFCN when the correct DAG is used, whereas all other approaches fail to yield good estimates. At best, when a partially correct graph is used, such as DAG 2 in the simulation ('partially' insofar as it more closely aligns with the true DGP than DAG 1 in terms of the relationships crucial for estimating the target ATE), the results are closer than with no DAG at all, which is the case for the typical machine learning approaches seen with the use of an random forest, transformer, or MLP. The key takeaway from this is that specifying a graph, even if it is incorrect, provides a transparent means to represent

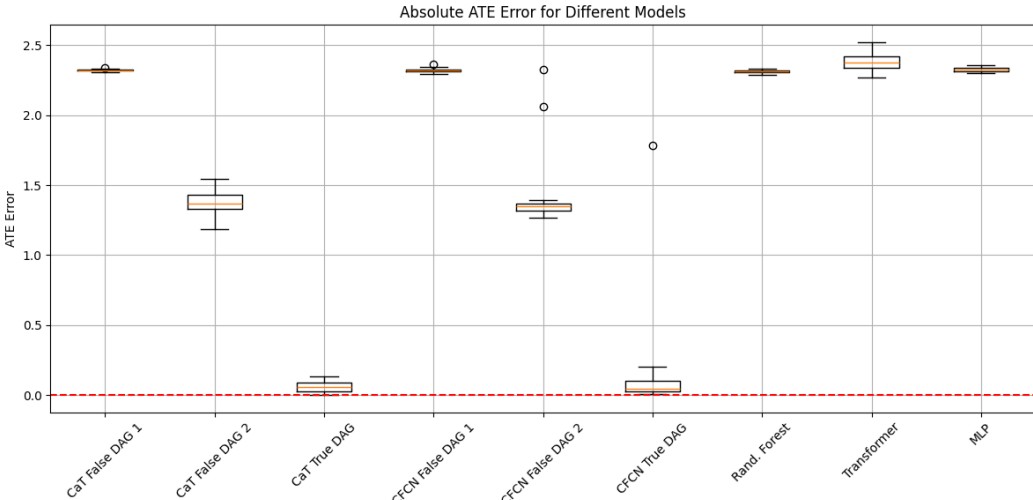

Figure 6: Illustrating the absolute error on the Average Treatment Effect estimation associated with conventional machine learning models, random forest (Breiman, 2001) (RF), multilayer perceptron (MLP), a transformer (Vaswani et al., 2017), and our CFCN and CaT networks. We show the results for CFCN and CaT under three different graphs (one correct, and two false).

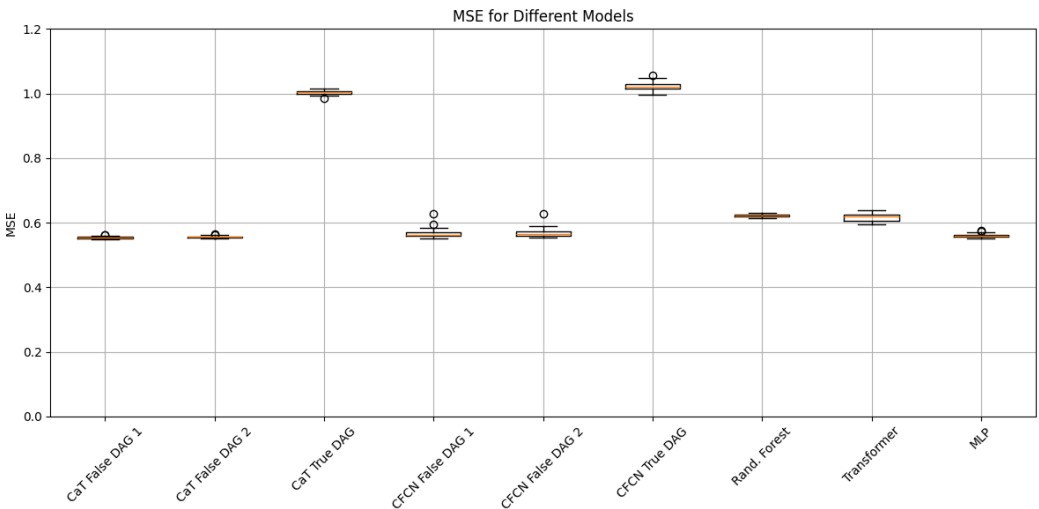

Figure 7: Illustrating the out-of-sample mean squared error for the main outcome $Y$ associated with conventional machine learning models, random forest (Breiman, 2001) (RF), multilayer perceptron (MLP), a transformer (Vaswani et al., 2017), and our CFCN and CaT networks. We show the results for CFCN and CaT under three different graphs (one correct, and two false). One can see that the correct causal graph has the highest MSE, whereas under all other conditions / with all other approaches the MSEs are comparable.

the assumptions concerning the underlying DGP, and may yield more meaningful results than failing to specify any constraints at all.

Finally, the complete results for the MSE on the estimation on the outcome variable $Y$ in the validation set are shown in Fig. 7. One can see that the use of the correct causal graph actually results in the worst estimation performance (despite having the best robustness and ATE estimation performance).

## S11. Causal Inference Evaluation

### S0.1 Jobs

Following descriptions in (LaLonde, 1986; Vowels et al., 2021b; Louizos et al., 2017a; Shalit et al., 2017) we also use the job outcomes dataset (*Jobs*) (LaLonde, 1986; Smith & Todd, 2005) (available at `https://users.nber.org/~rdehejia/data/.nswdata2.html`). Unlike the IHDP dataset, Jobs is real-world data with a binary outcome. We follow a similar approach to (Shalit et al., 2017), who used the Dehejia-Wahba (Dehejia & Wahba, 2002) sample along with a PSID comparison group. This dataset includes 260 treated samples and 185 control samples, in addition to the 2,490 samples from the PSID comparison group. It combines both observational and randomized control trial (RCT) data. Following the methodology of (Louizos et al., 2017a; Shalit et al., 2017), we apply a 56/24/20 train/validation/test split and conduct 100 runs with different random split assignments to estimate the average performance and standard error. Note that we use the same random seed across models for both initialization and dataset splitting, ensuring that the variance due to these factors is consistent across experiments. As in (Louizos et al., 2017a; Shalit et al., 2017; Yao et al., 2018), we evaluate our network on the Jobs dataset (where only partial effect supervision is available) by measuring the Average Treatment Effect on the Treated (ATT) error:

$$
\begin{aligned}
ATT = {} & ||D_1|^{-1} \sum_{i \in D_1} Y_i - |D_0|^{-1} \sum_{j \in D_0} Y_j \\
& - |D_1|^{-1} \sum_{i \in D_1} \hat{Q}(1, \mathbf{L}_{\text{set},i}) - \hat{Q}(0, \mathbf{L}_{\text{set},i})
\end{aligned}
\tag{14}
$$

where, in a slight abuse of notation $D = D_1 \cup D_0$ constitutes all individuals in the RCT, and the subscripts denote whether or not those individuals were in the treatment (subscript 1) or control groups (subscript 0). The first two terms in Eq. 14 comprise the true ATT, and the third term the estimated ATT. We may use the policy risk as a proxy for the Precision in the Estimation of the Heterogeneous Effect (PEHE - see Eq. 18):

$$
\begin{aligned}
\mathcal{R}_{pol} = {} & 1 - \mathbb{E}_{D1}[Y(1)|\pi(\mathbf{L}_{\text{set}}) = 1] \cdot p_D(\pi(\mathbf{L}_{\text{set}}) = 1) \\
& - \mathbb{E}_{D0}[Y(0)|\pi(\mathbf{L}_{\text{set}}) = 0] \cdot p_D(\pi(\mathbf{L}_{\text{set}}) = 0)
\end{aligned}
\tag{15}
$$

where $\pi(\mathbf{L}_{\text{set},i}) = 1$ is the policy to treat when $\hat{Y}_i(1) - \hat{Y}_i(0) > \alpha$, and $\pi(\mathbf{L}_{\text{set},i}) = 0$ is the policy not to treat otherwise (Yao et al., 2018; Shalit et al., 2017). $\alpha$ is a treatment threshold. This threshold can be varied to understand how treatment inclusion rates affect the policy risk. We set $\alpha = 0$, as per (Shalit et al., 2017; Louizos et al., 2017a). The subscripts for the expectation operator $\mathbb{E}$ and probability $p$ indicate the specific population to which these notations apply.

### S0.2 Twins

We access the Twins (Almond et al., 2005) dataset from this package / repository: `https://github.com/bradyneal/realcause` The dataset is based on births of twins in the USA between 1989 and 1991. We designate the heavier twin as the treatment group (t = 1) and the lighter twin as the control (t = 0). The outcome measured is 1-year mortality. For each pair of twins, 30 features were collected, including details about the parents, the pregnancy, and the birth, such as marital status, race, residence, previous births, pregnancy risk factors, care quality during pregnancy, and the number of gestational weeks.

For the Twins dataset, we provide an estimate for the average error on the estimated Average Treatment Effect (eATE) both within- and out-of-sample:

$$
\text{eATE} = |\tau - \hat{\tau}|
\tag{16}
$$

or

$$
\text{eATE} = \left| \tau - \frac{1}{n} \sum_{i=1}^{n} \left( \hat{Y}_1(i) - \hat{Y}_0(i) \right) \right|
\tag{17}
$$

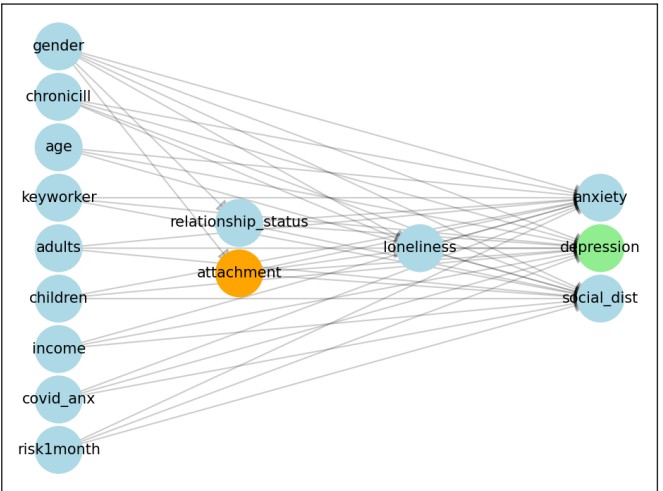

Figure 8: The DAG used in the real-world psychology example - reconstructed from the causal discovery and domain expertise results presented in (Vowels et al., 2023a). Treatment is attachment style 'attachment' (also highlighted in orange) and the two outcomes of interest at the measures of depression (highlighted in green).

where $\tau$ is the true average treatment effect and $\hat{Y}_1$ and $\hat{Y}_0$ are the estimated potential outcomes under treatment and no treatment, respectively.

Table 6: Number of items / dimensionality of each construct used in the real-world data analysis.

| Construct | Dimensionality |
|---|---|
| Attachment Style | 1 |
| Loneliness | 3 |
| Chronic Illness | 1 |
| Relationship Status | 1 |
| Social Distancing | 16 |
| Depression | 9 |
| Anxiety | 7 |
| Age | 1 |
| Gender | 1 |
| Keyworker Status | 1 |
| Adults in Household | 1 |
| Children in Household | 1 |
| Change in Income | 1 |
| Covid Anxiety | 1 |
| Risk Taking in the Past Month | 1 |

## S0.3   REAL-WORLD DATASET

We performed a secondary analysis of UK data from wave 2 of the COVID-19 Psychological Research Consortium Study (C19PRCS), a longitudinal, internet-based survey. Comprehensive details about the methodology can be found in (McBride et al., 2021)), and the data are openly available on the OSF (https://osf.io/v2zur/). In brief, the UK portion of the C19PRC Study was conducted in April/May 2020 for Wave 2. During this wave, strict social distancing measures were

enforced. Quota sampling ensured the recruitment of a nationally representative panel of UK adults, based on age, gender, and household income. Participants had to be at least 18 years old, UK residents, and capable of completing the survey in English. After giving consent, participants completed the survey online and were compensated by Qualtrics for their time. Ethical approval was granted by a UK university psychology department (Reference: 033759). In Wave 2, 1406 individuals took part, but participants who did not report their attachment styles were excluded from the analysis, and after listwise deletion of rows with other missing values, we arrived at a sample size for this wave of 895 participants.

The motivation for these particular analyses of these data stems from how the COVID-19 pandemic posed numerous challenges, including balancing health protection with maintaining a fulfilling life. For some individuals, the mental health impact of the pandemic has been more severe than for others. The original study aimed to build a theoretical causal model to better understand how individual differences in attachment styles affect both social distancing compliance and mental health outcomes (loneliness, depression, and anxiety) during the pandemic.

| | Twins | | | Jobs | | |
|---|---|---|---|---|---|---|
| Model | eATE ws | eATE os | Risk ws | Risk os | eATT ws | eATT os |
| GANITE | .0058 ± .0017 | .0089 ± .0075 | .13 ± .01 | .14 ± .00 | .01 ± .01 | .06 ± .03 |
| CFRwass | .0112 ± .0016 | .0284 ± .0032 | .17 ± .00 | .21 ± .00 | .04 ± .01 | .09 ± .03 |
| Transformer | .0078 ± .0056 | .0072 ± .0085 | .2352 ± .0171 | .2505 ± .0612 | .0165 ± .0118 | .0813 ± .0585 |
| Random Forest | .0071 ± .0032 | .0084 ± .0061 | .0704 ± .0144 | .2518 ± .0553 | .0337 ± .0164 | .0909 ± .0654 |

Table 7: Results ± standard deviations for the Twins and Jobs benchmark datasets comparing with non-causal estimators. eATE - absolute error on the estimation of the Average Treatment Effect; Risk - Policy risk; eATT - absolute error on the estimation of the Average Treatment effect on the Treated; ws - within sample; os - out of sample. Methods included are GANITE (Yoon et al., 2018), CFRWass (Shalit et al., 2017), a simple transformer (Vaswani et al., 2017), and a random forest (Breiman, 2001).

We follow closely the process in (Vowels et al., 2023a) for estimating the causal effect of shifting from one category of attachment style to another on depression. We also report the results for a subset of their analyses in Table 3, which use a 'naive' estimator (comprising the bivariate linear model between the categorical treatment 'attachment' and the two outcomes), a targeted learning estimator specialized for causal inference which incorporates semi-parametric techniques (van der Laan & Starmans, 2014; Vowels et al., 2023b), and our results using CaT. Note that there may be some minor differences in their data preprocessing which we were not able to reproduce. In particular, for each node in the DAG, the original authors reduced the dimensionality of the construct to be uni-dimensional by taking the sum of the scores for each of the individual items. In contrast, we padded all input variables so that they were the same dimensionality as the node with the highest dimensionality. For instance, social distancing 'social dist' was found to have 16 items, so loneliness, which has only 3 items, was zero-padded to have 16 dimensions. The enables us to use all available information in the input. The dimensionalities / number of items for each construct are shown in Table reftab:realworlddimensions. We also use the DAG presented in (Vowels et al., 2023a) which was the result of a causal discovery process alongside domain expertise, this DAG is reproduced in Figure 8.

For estimating the causal effect of a change in attachment style, we treat secure attachment as the reference group, and the three other categories (fearful, anxious, and avoidant) as the comparison categories. We estimate the effect of changing groups by setting the value of attachment to each of the 4 categories and record the shift in depression. Note the anxiety measure has 7-items (GAD-7; (Spitzer et al., 2006)) and the depression measure has 9 (PHQ-9; (Kroenke et al., 2002)) we take the average over the 9 dimensions to estimate the final effect on depression.

Whilst standard errors can be readily obtained from the targeted learning ('TL') and bivariate linear modeling ('naive') approaches, we used boostrapping to obtain estimates of the standard errors for the estimates from CaT. To do this, we repeated the estimation process 100 times using a sampling-with-replacement process and a sampling fraction of 0.9 of the full dataset.

In Table 3 we see that, despite CaT not being specialized to causal inference, the results are reasonably close to those of the TL estimator. Note that whilst our method uses the full dimensionality of each construct, the comparison methods (naive and TL) use the aggregated scales - they have no other option but to do so. The standard errors are notably wider, which may be related to the use of bootstrapping for deriving them; in contrast to TL, which can be fit once and used to derive standard errors, the CaT is initialized and optimized from fresh for each bootstrap simulation, and trained on a smaller subsample of the full dataset than TL. As such, we expect that this method for estimating the effect size would yield less tight intervals than TL. Furthermore, and as mentioned in the limitations section of the main text, the presence of 'loneliness' in the graph maybe also contribute estimation variance as the recursive inference process has to provide intermediate predictions for this mediator before using these predictions to, in turn, predict the outcome. Finally, the TL estimator includes the use of a Super Learner (van der Laan et al., 2007), which involves a cross-validation approach to finding the optimal set of candidate learners for the plug-in-estimator prediction task, whereas we did not undertake any equivalent form of hyperparameter search for CaT, making TL a challenging approach to compete with. Regardless, it is difficult to make a comparison with this real-world application, because no ground truth is available.

| Model | PEHE ws | PEHE os |
|---|---|---|
| RF (Breiman, 2001) | 1.84 ± 0.08 | 1.79 ± 0.03 |
| CT (Athey & Imbens, 2016) | 4.81 ± 0.18 | 4.96 ± 0.21 |
| CF (Wager & Athey, 2018) | 2.16 ± 0.17 | 2.18 ± 0.19 |
| BART (Chipman et al., 2010) | 2.13 ± 0.18 | 2.17 ± 0.15 |
| CFR (Shalit et al., 2017) | 2.05 ± 0.18 | 2.18 ± 0.20 |
| DR-CFR (Hassanpour & Greiner, 2020) | 2.44 ± 0.20 | 2.56 ± 0.21 |
| GANITE (Yoon et al., 2018) | 2.78 ± 0.56 | 2.84 ± 0.61 |
| CEVAE (Louizos et al., 2017a) | 3.12 ± 0.28 | 3.28 ± 0.35 |
| TEDVAE (Zhang et al., 2021) | 1.75 ± 0.14 | 1.77 ± 0.17 |
| CaT | 4.22 ± 0.25 | 4.26 ± 0.23 |
| CFCN | 4.14 ± 0.13 | 4.17 ± 0.10 |

Table 8: Comparison of methods based on within-sample (ws) PEHE and out-of-sample (os) PEHE for the ACIC 2016 causal inference challenge (Dorie, 2016) including results presented in (Zhang et al., 2021).

## S12. A CURIOUS RESULT

Table 7 presents a curious result for how the same scikit-learn (Pedregosa et al., 2011) random forest (RF) (Breiman, 2001) and transformer (Vaswani et al., 2017) used in the motivating experiments perform on the two causal inference benchmarks used in this work. Specifically, the RF and transformer (using default/untuned hyperparameters) perform very competitively with a set of methods specifically designed for causal inference. Further work is required to explore this result, but it echos a message from (Curth et al., 2021) that these benchmarks for causal inference should be reconsidered.

We also provide results for the Atlantic Causal Inference Challenge (ACIC) 2016 (Dorie, 2016) which is available here: `https://github.com/vdorie/aciccomp/tree/master/2016` and follow the procedure and present relevant results from (Zhang et al., 2021). This dataset comprises 77 different data generating processes, each with 58 variables and 4802 observations. We undertake a 60/30/10% train/validation/test split, although we discard the 30% validation set for hyperparameter tuning as such tuning is unrealistic in real-world causal inference. The results for the Precision in the Estimation of the Heterogeneous Effect (PEHE) are given in Table 8. This table include results from the Causal Tree (CT; (Athey & Imbens, 2016)), Disentangled Counterfactual Regression (DR-CFR; (Hassanpour & Greiner, 2020) and Treatment Effect Estimation with Disentangled Variational Autoencoder (TEDVAE; (Zhang et al., 2021)). The PEHE is defined as:

$$\epsilon_{\text{PEHE}} = \sqrt{n^{-1} \sum_{i=1}^{n} (\tau_i - \hat{\tau}_i)^2}, \tag{18}$$

where $\hat{\tau}_i$ is the effect for observation $i$ from the dataset estimated by the model.

For these results, the hyperparameters for CaT were: the number of layers $n_{\text{layers}} = 3$, input embedding dimension = 20, head size = 10, number of attention heads = 5, feed-forward embedding / projection dimension = 10, and dropout = 0.0. We used the same hyperparameters for CFCN as in Table 5.

Once again, the RF also performs well on the ACIC 2016 benchmark as can be seen from Table 8. As successful causal inference requires one to know the underlying DAG, the causal inference benchmarks make unbiased estimation possible with a set of default / traditional estimators (like the random forest), because all of them assume the same graph (i.e. one treatment variable, a set of possible confounders or precision variables, and an outcome). At least based on this preliminary result, we recommend researchers to at least double-check the performance of standard estimators on these evaluation datasets, to validate that progress is indeed being made.

Future benchmarks might, for instance, include an element of causal discovery or causal feature selection, in order to really distinguish the challenge of causal inference from a regular supervised learning task. One of the advantages of CaT and CFCN is that it makes the inference question flexible - one can estimate the causal effect between any pair of nodes in the graph. Furthermore, this graph is specified explicitly and could, for example, represent the output of upstream causal discovery.

Based on these results, we are not confident that causal inference benchmark datasets are really highlighting the key performance differences between proposed methods. We once again re-iterate that CaT and CFCN are not tuned for causal inference, although this represents a promising direction. When evaluating CaT and CFCN on standard causal inference benchmarks which assume a basic DAG structure constituting a set of covariates, a treatment, and an outcome, the estimation problem is reduced to a simple choice of estimator, and using CaT is similar to using a typical transformer.

## S13. THEORETICAL FOUNDATION FOR CAT'S ARCHITECTURE

Theorem 1 shows that, under our architectural assumptions, CaT defines a Causal Bayesian Network which has conditional factorization which matches that implied by the supplied DAG and that interventions implemented through the masking and embedding mechanism coincide with the standard g–formula. Theorem 2 and Corollary 1 formalize the covariate shift and transportability claims: if the causal mechanisms are invariant across domains, CaT's interventional queries (such as ATE) are invariant across domains, and predictions for a target node are provably insensitive to shifts in variables outside its ancestor set.

### ARCHITECTURAL ASSUMPTIONS AND CAUSAL SEMANTICS

**Review of terminology and notation.** The tensor $\mathbf{X} \in \mathbb{R}^{B \times |Z| \times C}$ contains, for each mini-batch element, the $C$-dimensional raw features for every node in the DAG. The slice $\mathbf{X}_i = \mathbf{X}[:, i, :]$ is the raw input for node $i \in Z$. Embeddings are produced by $|Z|$ independent linear maps to give $\mathbf{X}_E \in \mathbb{R}^{B \times |Z| \times d_E}$ where $d_E \geq C$ and $d_E > 1$ (required for disentanglement). The learnable query embedding is $\boldsymbol{\gamma} \in \mathbb{R}^{|Z| \times d_E}$. The topologically sorted adjacency is $\mathbf{A} \in \{0, 1\}^{|Z| \times |Z|}$ with $\mathbf{A}_{ji} = 1$ if and only if $j \in \text{pa}(i)$.

**Cross-attention structure.** Unlike standard self-attention, CaT employs causally-masked cross-attention (CMCA) where queries $\mathbf{Q} = \boldsymbol{\gamma} \mathbf{W}_Q + \mathbf{b}_Q$ derive from the learnable embedding $\boldsymbol{\gamma}$ in the first block (then from previous block outputs), while keys $\mathbf{K} = \mathbf{X}_E \mathbf{W}_K + \mathbf{b}_K$ and values $\mathbf{V} = \mathbf{X}_E \mathbf{W}_V + \mathbf{b}_V$ always derive from the embedded input $\mathbf{X}_E$. This enables iterative refinement by extracting parental information at each layer.

**Assumption 1** (Hard parental mask)**.** At every block, attention from token $i$ can only read from its parents $\text{pa}(i)$. This is enforced through

$$\mathbf{O} = \text{softmax}\left(\mathbf{A}^\top \circ \frac{\mathbf{Q}\mathbf{K}^\top}{\sqrt{h_s}}\right) \cdot \mathbf{V},$$

where $(\mathbf{A}^\top)_{ij} = 1$ if and only if $j \in \mathrm{pa}(i)$. Since $\mathbf{A}$ has zeros on the diagonal (no self-loops), $(\mathbf{A}^\top)_{ii} = 0$, preventing self-attention. Thus attention weights from any $S \subseteq Z \setminus \mathrm{pa}(i)$ to $i$ are zero.

**Assumption 2** (Per-node embedding). The embedding stage uses $|Z|$ independent linear layers, so the embedded token $\mathbf{X}_{E,i}$ depends only on $\mathbf{X}_i$. This preserves causal independence at the input level and enables heterogeneous feature dimensions across nodes.

**Assumption 3** (Tokenwise operations). Residual connections and the positionwise feed-forward $FF(\cdot)$ act independently across tokens. Each CBlock preserves this structure:

$$\mathbf{O}_{\mathrm{CMCA}}^{r,'} = \mathrm{CMCA}(\mathbf{O}_{\mathrm{Block}}^{r-1}, \mathbf{X}_E) + \mathbf{O}_{\mathrm{Block}}^{r-1}, \tag{19}$$

$$\mathbf{O}_{\mathrm{Block}}^r = FF(\mathbf{O}_{\mathrm{CMCA}}^{r,'}) + \mathbf{O}_{\mathrm{CMCA}}^{r,'}, \tag{20}$$

where $\mathbf{O}_{\mathrm{Block}}^0 = \boldsymbol{\gamma}$ and batch normalization is optional. No layer normalization is used as it would rescale potentially calibrated interventions.

LAYERWISE DEPENDENCE AND LOCAL MARKOV PROPERTY

Let $h_i^{(\ell)}$ denote token $i$'s hidden state after block $\ell$; by construction $h_i^{(L)}$ feeds the prediction head for node $i$.

**Proposition 1** (Parental sufficiency). *Under Assumptions 1–3, for any block $\ell \geq 1$ and node $i \in Z$, there exists a measurable function $F_i^{(\ell)}$ such that*

$$h_i^{(\ell)} = F_i^{(\ell)}(\mathbf{X}_{\mathrm{pa}(i)}).$$

*Consequently, for any $S \subseteq Z \setminus \mathrm{pa}(i)$,*

$$h_i^{(\ell)} \perp\!\!\!\perp \mathbf{X}_S \mid \mathbf{X}_{\mathrm{pa}(i)}.$$

**Intuition.** Token $i$ aggregates only from its parents through causally-masked cross-attention and cannot attend to itself due to the zero diagonal in $\mathbf{A}$. Since $\mathbf{X}_E$ is fed at every layer, each block allows the evolving query (from $\mathbf{O}_{\mathrm{Block}}^{r-1}$) to extract relevant parental information (keys/values from $\mathbf{X}_E$). The residual connections maintain and refine token $i$'s representation across layers without requiring self-attention. The hidden state at $i$ thus depends only on parental inputs $\mathbf{X}_{\mathrm{pa}(i)}$.

*Remark* 1 (Strict causal dependency through masking). The architecture enforces strict causal dependency: token $i$ never directly accesses $\mathbf{X}_i$ through attention. The DAG mask with zero diagonal ensures $h_i^{(\ell)}$ is constructed purely from parental information.

Let $\widehat{\mathbf{X}}_i$ be the output of the final linear prediction head for node $i$, computed from $h_i^{(L)} = \mathbf{O}_{\mathrm{Block},i}^L$ after $L$ CBlocks.

**Lemma 1** (Local Markov property). *Under Assumptions 1–3, with the final linear layer acting independently per token,*

$$\widehat{\mathbf{X}}_i = g_i(\mathbf{O}_{Block,i}^L) = g_i(F_i^{(L)}(\mathbf{X}_{\mathrm{pa}(i)})).$$

*Since the mask structurally prevents self-attention, the prediction is necessarily a function of $\mathbf{X}_{\mathrm{pa}(i)}$ alone, yielding*

$$p_{CaT}(x_i \mid x_{Z \setminus \{i\}}) = p_{CaT}(x_i \mid x_{\mathrm{pa}(i)}).$$

**Intuition.** The final prediction head reads token $i$'s representation after all CBlocks. Since the DAG mask has zero diagonal, token $i$ never accesses $\mathbf{X}_i$ during forward passes. This architectural constraint enforces the local Markov property of a Causal Bayesian Network with respect to $\mathcal{G}$.

**Theorem 1** (Structural soundness and DAG factorization). *Under Assumptions 1–3, the joint predictive distribution defined by CaT factorizes according to the DAG $\mathcal{G}$:*

$$p_{CaT}(x_1, \ldots, x_{|Z|}) = \prod_{i \in Z} p_{CaT}(x_i \mid x_{\mathrm{pa}(i)}).$$

*In particular, CaT implements a Causal Bayesian Network over $Z$ whose conditional distributions respect the parental sets in $\mathcal{G}$.*

**Intuition.** The CMCA mask guarantees that each node's representation depends only on its parents. The parallel attention computation thus yields a model that satisfies the local Markov property for all nodes. The standard result from graphical models then implies the global factorization with respect to $\mathcal{G}$.

IDENTIFIABLE INTERVENTIONS VIA MASKED ATTENTION

We now present how CaT implements the truncated factorization (g-formula) used in causal inference (Robins, 1986).

**Definition 1.** For an intervention set $\mathcal{I} = \{(\mu, z) : z \in S\}$ with $S \subseteq Z$, CaT implements $do(X_S = \mu)$ by:

1. Replacing embeddings for intervened nodes: $\mathbf{X}^{do}_{E,j} = \text{Embed}_j(\mu)$ for $j \in S$.

2. Processing $\mathbf{X}^{do}_E$ through all CBlocks with unchanged CMCA masks.

3. Reading outputs where intervened nodes retain their set values:

$$X_j^{do} = \begin{cases} \mu, & j \in S, \\ \widehat{X}_j(\mathbf{O}^L_{\text{Block},j}), & j \notin S. \end{cases}$$

Non-intervened nodes automatically incorporate upstream interventions since their representations depend only on (possibly intervened) parents.

**Proposition 2** (Truncated product via CMCA). *Under Theorem 1, the CMCA architecture with Definition 1 directly implements the g-formula (Robins, 1986)*

$$p_{CaT}\big(x_Z \mid do(x_S)\big) = \prod_{j \in Z \setminus S} p_{CaT}\big(x_j \mid x_{\text{pa}(j)}\big)\Big|_{x_{\text{pa}(j)} \leftarrow x^{do}_{\text{pa}(j)}}.$$

*Treatment effects like $\boldsymbol{\tau} := \mathbb{E}\big[Y \mid do(D{=}\mu_1)\big] - \mathbb{E}\big[Y \mid do(D{=}\mu_0)\big]$ are therefore computed by forward passes with different intervention values.*

**Intuition.** Interventions work naturally in CaT: setting $\mathbf{X}_{E,j}$ to intervention values affects only downstream nodes (children of $j$), since node $j$ itself cannot self-attend. The mask ensures each node reads only from parents, so intervention effects propagate causally downstream in a single forward pass. The resulting interventional distribution coincides with the truncated product of a Causal Bayesian Network over $\mathcal{G}$.

COVARIATE SHIFT, STRUCTURAL ROBUSTNESS, AND TRANSPORTABILITY

We now formalize how CaT behaves under covariate shift and connect this to standard assumptions in causal transportability. The key result is a *structural robustness* property: assuming that the DAG is correct and the causal mechanisms are invariant, CaT's predictions do not change when covariates that are not causal ancestors of the target shift between domains. The usual transportability of interventional queries then follows as a consequence.

Consider two domains (or environments) $e \in \{s, t\}$ with the same DAG $\mathcal{G}$ and the same structural conditionals

$$p^{(e)}\big(x_i \mid x_{\text{pa}(i)}\big) = p\big(x_i \mid x_{\text{pa}(i)}\big) \quad \text{for all } i \in Z,$$

but possibly different observational distributions $p^{(e)}(x_Z)$ due to different exogenous noise or covariate distributions.

**Assumption 4** (Consistent estimation of conditional distributions). CaT is trained in domain $s$ with sufficient capacity and data so that for all $i \in Z$,

$$p^{(s)}_{\text{CaT}}\big(x_i \mid x_{\text{pa}(i)}\big) \xrightarrow{N \to \infty} p\big(x_i \mid x_{\text{pa}(i)}\big).$$

The same architecture is then applied in domain $t$ without retraining.

**Theorem 2** (Structural robustness under covariate shift). *Let $Y$ be a target node, and suppose domain shift affects only a subset $S \subseteq Z \setminus \mathrm{An}(Y)$ that contains no ancestors of $Y$. Under Assumptions 1–3 and 4, for any fixed configuration of parental variables $x_{\mathrm{pa}(Y)}$, the CaT predictor for $Y$ is invariant across such covariate shift:*

$$p_{CaT}^{(s)}\big(y \mid x_{\mathrm{pa}(Y)}\big) \;=\; p_{CaT}^{(t)}\big(y \mid x_{\mathrm{pa}(Y)}\big) \quad \textit{for all } y, x_{\mathrm{pa}(Y)}.$$

*In other words, CaT is robust to covariate shift in variables that are not causal ancestors of $Y$.*

**Intuition.** Due to the CMCA mask, the representation for $Y$ can only depend on its parents and their ancestors. Variables outside $\mathrm{An}(Y)$ never influence $Y$'s token representation. In contrast, non-causal transformers/networks are vulnerable to this: in a generic neural network, the representation of $Y$ is free to depend on any input variable, so non-ancestors can spuriously influence predictions and make them sensitive to distributional changes that do not alter the underlying mechanisms.

**Corollary 1** (Transportability of identified causal queries). *Under Assumptions 1–3 and 4, any causal query that is identified by the g-formula (Robins, 1986) in $\mathcal{G}$, such as the average treatment effect*

$$\tau = \mathbb{E}\big[Y \mid do(D{=}\mu_1)\big] - \mathbb{E}\big[Y \mid do(D{=}\mu_0)\big],$$

*has a CaT estimate that is invariant across domains $s$ and $t$ in the large-sample limit:*

$$\tau_{CaT}^{(s)} \xrightarrow{N \to \infty} \tau \quad \textit{and} \quad \tau_{CaT}^{(t)} \xrightarrow{N \to \infty} \tau.$$

*Equivalently, as long as the mechanisms $p(x_i \mid x_{\mathrm{pa}(i)})$ are shared across domains, CaT's interventional estimates are transportable.*

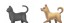

