# OpenReview forum: "CaTs and DAGs: Integrating Directed Acyclic Graphs with Transformers for Causally Constrained Predictions"
_ICLR.cc/2026/Conference — ICLR 2026 Poster_

### Official Review · Reviewer_5yaS · 2025-10-19

**Soundness:** 4
**Presentation:** 2
**Contribution:** 2
**Rating:** 4
**Confidence:** 4

**Summary:**

This paper introduces causal transformers (CaTs) for causal prediction. CaTs encode the information of a given causal graph in the architecture, offering a strong approximation power for a given causal model. Detailed experiments show the superior performance of CaTs in term of ATE estimation.

**Strengths:**

1. A key advantage highlighted is that by constraining the model to a (correct) causal DAG, it learns to ignore spurious correlations in the training data. This makes the model more robust and stable when faced with distributional shifts (i.e., covariate shift), a common failure point for traditional machine learning models.

2. The experiments show superior performance over the traditional random forest models even if the DAG is misspecified.

3. Unlike many causal inference methods designed for single-value, tabular data, the Causal Transformer (CaT) architecture is built to handle high-dimensional inputs. It can accept a sequence of embeddings, making it applicable to complex, multimodal data like text, images, or multi-item questionnaires.

**Weaknesses:**

1. The literature review part is not complete. There have been many works in the literature that consider using neural networks for counterfactual estimation. For instance, [1] establishes the connection between neural nets and causal models and constructs feed-forward networks for discrete causal models. [2] establishes similar results for causal models with mixed variables (continuous and discrete) and uses it for partial identification. Similar to this work, [3] proposes causal transformers for counterfactual estimation. There are many other works on this topic as well. Please include those works in the related work part.

2. Some details are unclear in the paper. Please see the question part.

3. In the experiment, the author compared the ATE given by different models. However, it is well-known that the Double Robust (DR) estimator provides a more robust way for estimation. It will provide more practical implications if the author can provide a comparison (using the same propensity model).

[1] Xia, Kevin, et al. "The causal-neural connection: Expressiveness, learnability, and inference." Advances in Neural Information Processing Systems 34 (2021): 10823-10836.

[2] Tan, Jiyuan, Jose Blanchet, and Vasilis Syrgkanis. "Consistency of neural causal partial identification." Advances in Neural Information Processing Systems 37 (2024): 68956-68999.

[3] Melnychuk, Valentyn, Dennis Frauen, and Stefan Feuerriegel. "Causal transformer for estimating counterfactual outcomes." International conference on machine learning. PMLR, 2022.

**Questions:**

Clarify questions:

1. Does CaTs apply to all kinds of causal graphs or just the graph in Figure 2? If it applies to all causal graphs, the identification formula (1) seems to be incorrect since $L_{\text{set}}$ may contain colliders.

2. How do you model hidden confounders in Figure 3 (b)?

3. Can the authors explain the use of learnable embedding $\gamma$? In particular, why does $Q$ depend on $\gamma$ but $K,V$ do not?

4. The authors mention that they do not include normalization in CaTs because it may impact the capacity for precise estimation. In a standard transformer, LayerNorm is applied to each variable (token) independently across its feature dimension. It does not typically "contaminate" information between variables. Can the authors provide a more detailed explanation?

Experiment questions:

5. The authors include a standard transformer in Table 1, but do not include it Table 2. Can you include the results too?

6. In Table 1, can you explain why misspecified DAGs can yield low MSE? This is quite counterintuitive.

---

> ### Author Response · Authors · 2025-11-20
> **Response to Reviewer 5yaS Part 1/N**
>
> We thank the reviewer for their consideration, time, and constructive feedback, as well as highlighting the increased robustness, superior performance of our method above misspecified alternatives without the possibility of causal constraint, and the flexibility that our model has in handling high-dimensional inputs. We respond to each concern in turn below.
>
> **Weakness 1:** *The literature review part is not complete. There have been many works in the literature that consider using neural networks for counterfactual estimation. For instance, [1] establishes the connection between neural nets and causal models and constructs feed-forward networks for discrete causal models. [2] establishes similar results for causal models with mixed variables (continuous and discrete) and uses it for partial identification. Similar to this work, [3] proposes causal transformers for counterfactual estimation. There are many other works on this topic as well. Please include those works in the related work part.*
>
> We totally agree with the reviewer that counterfactual estimation with neural networks is a large domain. We tried to capture some of this literature already; we include 14 such references and evaluate against 4 of these directly. Regardless, we thank the reviewer for the three references. Note that [3] is already part of our literature review (paragraph 4 of Section 2), and [2] is predominantly a proposal for an algorithm for causal identification paired with estimation (rather than a general architectural proposition), but we have included [1] and [2] in addition.
>
> ---
>
> **Weakness 2 – see questions below for response to this.**
>
> ---
>
> **Weakness 3:** *In the experiment, the author compared the ATE given by different models. However, it is well-known that the Double Robust (DR) estimator provides a more robust way for estimation. It will provide more practical implications if the author can provide a comparison (using the same propensity model).*
>
> We thank the reviewer for the comment and agree that DR methods are worth discussing. Firstly, we note that DR estimators are targeted to specific estimands. The CaT framework is general in that it can be used to estimate the average causal effect of any identifiable intervention on any node in the DAG. If one wanted to integrate CaT into a DR framework, one would have to either prioritise a specific estimand, or apply the framework to each estimand of interest.
>
> Secondly, CaT would otherwise fit naturally as either an outcome or a propensity model within DR and, as such, we would argue that DR methods are somewhat orthogonal to the aims of our work. DR approaches in general require substitution or plug-in statistical learning algorithms (of which CaT is an example). Whilst we have not attempted a direct adaptation of CaT to make it DR, both the TVAE method in Table 2 and the TL (Targeted Learning) method used for the psychological data are doubly robust estimators, so a comparison of performance is already present. Our positioning is such that our evaluation already provides a comparison of CaTs against both DR and non-DR methods, and the approach itself is agnostic to the choice of statistical learner, leaving users the choice as to whether to integrate the method into a DR framework or not. It is entirely possible, for instance, to treat CaT as a statistical learning algorithm, and to update its estimates for average treatment effect using a propensity score model following the targeted learning framework.
>
> ---
>
> **Question 1:** *Does CaTs apply to all kinds of causal graphs or just the graph in Figure 2? If it applies to all causal graphs, the identification formula (1) seems to be incorrect since $L_{set}$ may contain colliders.*
>
> We understand the reviewer's concern with equation 1, because based on the current notation, $L_{set}$ is indeed misleading or incorrect, as the conditioning set may contain mediators or colliders as the reviewer points out. For equation 1 we have changed this to $L_{sub}$ and defined this as a sufficient adjustment set.
>
> To answer the reviewer's question directly, the answer is yes: CaTs can be used to estimate the CATE or ATE (and also quantile effects) of *any identifiable query on any DAG*. Indeed, multiple DAGs were used in our evaluations; see Figure 8 in the supplementary for a 15-node graph used for estimating the causal effect within a real-world application.
>
> As such, the architecture enforces the DAG at the level of information flow, preventing paths that would correspond to conditioning on inappropriate variables. At inference, we implement recursive substitution: set intervention values, then update only descendants while keeping non-descendants fixed. This matches the standard g-formula (Robins, 1986) simulation and enables effects on any subset of nodes to be propagated downstream without opening backdoor paths.

---

> > ### Author Response · Authors · 2025-11-20
> > **Response to Reviewer 5yaS Part 2/N**
> >
> > (question 1 continued):
> >
> > Concretely, the routine applies the intervention to $D'$, finds all descendants of the intervened nodes, and iterates through nodes in causal order, recomputing predictions and writing back only the current node’s column. Colliders present in the dataset do not harm identification because masked attention forbids their influence unless the DAG contains an incoming edge to the target. Mediators are handled correctly by simulating them under the intervention and feeding those simulated values into their children, which corresponds to recursive substitution rather than conditioning for adjustment.
> >
> > Two final clarifications: in the example DAG with a mediation structure $(X \to M \to Y)$, a confounder structure $(X \gets Q \to Y)$, and the collider structure $(X \to C \gets Y)$, we first compute $M' = M \mid do(X=1)$ and then predict $Y$ using its parents only, i.e., $Y \mid (M', Q)$. This is exactly what the code implements via masked routing, and it generalizes to any identifiable interventional query on any DAG.
> >
> > ---
> >
> > **Question 2:** *How do you model hidden confounders in Figure 3 (b)?*
> >
> > We thank the reviewer for the question. They are not modelled explicitly in CaT or CFCN. In spite of these hidden confounders, the target estimand is still identifiable from the observed data according to the $d$-separation rules. We simply include the endogeneity for completeness. Of course, if hidden confounding is present and no valid proxies or instruments exist, identification is not guaranteed. When valid proxies exist, they can be included as parents; otherwise, it may be possible to extend CaT with a latent parent token, or amortized variational regularization to create a truly generative model, but this is beyond the scope of the present paper and represents interesting future work.
> >
> > ---
> >
> > **Question 3:** *Can the authors explain the use of learnable embedding $\gamma$? In particular, why does $Q$ depend on $\gamma$ but $K$, $V$ do not?*
> >
> > The learnable embedding $\boldsymbol{\gamma}$ is a randomly initialized parameter of shape $|\mathbf{Z}| \times d_E$ that serves as the model's initial predictive state. Unlike standard transformers where queries come from input tokens, $\boldsymbol{\gamma}$ starts as a “blank slate” that learns through training to extract causally-relevant information from the input, according to the permitted interactions enforced by the DAG.
> >
> > In cross-attention, queries ($\mathbf{Q}$) represent what we are trying to predict, while keys ($\mathbf{K}$) and values ($\mathbf{V}$) represent the information source. This is coherent with the principal motivation for cross-attention.
> >
> > In subsequent blocks, the previous block's output replaces $\boldsymbol{\gamma}$ as the query, but $\mathbf{K}$ and $\mathbf{V}$ continue to be derived from $\mathbf{X}_E$. This allows the model to iteratively refine its predictions by comparing them against the observed input at each layer, while the DAG mask ensures predictions only attend to their causal parents. We believe there is an analogy with iterative message-passing where predictions query observations under causal constraints, although we have not explored this analogy more deeply.
> >
> > Note that it is possible to initialize $\boldsymbol{\gamma}$ as either a learnable parameter, or to keep it static. In our experiments we observed marginally better results when it is learnable, but as its purpose is to aggregate information according to the conditional structure determined by the DAG, we cannot think of a reason why it would need to be learnable in principle, other than to potentially help with the optimisation process.
> >
> > ---
> >
> > **Question 4:** *The authors mention that they do not include normalization in CaTs because it may impact the capacity for precise estimation. In a standard transformer, LayerNorm is applied to each variable (token) independently across its feature dimension. It does not typically “contaminate” information between variables. Can the authors provide a more detailed explanation?*
> >
> > We thank the reviewer for highlighting this. What we said is indeed inaccurate and we have corrected this. The motivation for avoiding LayerNorm is slightly different, as we also discuss with one of the other reviewers. Instead, what we meant to explain is that while LayerNorm can be applied without violating the structural causal constraints when used carefully (specifically, when applied across only the embedding dimension $C$ independently), we opted to exclude it due to concerns about calibration and effect size estimation, particularly under interventions.
> >
> > As causal inference requires not just correct conditional independence structure, but also accurate effect size estimation, LayerNorm is problematic because it alters the scale and location of representations in a data-dependent manner. Consider two key scenarios: (tbc...)

---

> > > ### Author Response · Authors · 2025-11-20
> > > **Response to Reviewer 5yaS Part 3/3**
> > >
> > > (Continuation for question 4):
> > > 1. **Intervention calibration.** When performing interventions via our recursive inference algorithm (Algorithm 1), we set intervention variables to specific values (e.g., $do(D=1)$ vs. $do(D=0)$). LayerNorm would transform these intervention values based on sample-dependent statistics, potentially distorting the intended magnitude. For instance, setting $D=1$ for all samples would result in different normalised values depending on the other features in each sample's embedding.
> > >
> > > 2. **Effect size interpretation.** ATEs are defined on the original scale of the outcome variable. LayerNorm applies affine transformations that vary across samples and layers, making it difficult to recover effect sizes on the interpretable original scale. While the relative ordering of effects might be preserved, the magnitude becomes confounded with the normalisation statistics.
> > >
> > > We have adjusted the explanation accordingly.
> > >
> > > ---
> > >
> > > **Question 5:** *The authors include a standard transformer in Table 1, but do not include it in Table 2. Can you include the results too?*
> > >
> > > These results are actually already in the supplementary material alongside a discussion about the appropriateness of causal benchmarks, but were moved because of space constraints. We will move these back into the main paper (CRC) if the paper is accepted.
> > >
> > > ---
> > >
> > > **Question 6:** *In Table 1, can you explain why misspecified DAGs can yield low MSE? This is quite counterintuitive.*
> > >
> > > This is a great question, and in our view the phenomenon can lead to significantly misleading results during model selection processes. Essentially, the inclusion of extra, non-parent edges can introduce shortcuts that correlate with the variable being predicted. This can reduce prediction error while biasing causal effects and increasing the model's sensitivity to covariate shift. Collider structures are a good example of this: colliders between cause $X$ and effect $Y$ are highly correlated with $Y$. As such, their inclusion (e.g., by virtue of miscategorising them as precision variables rather than colliders) can lead to lower MSE scores, even though the DAG is misspecified and the causal effect estimates are biased.

---

### Official Review · Reviewer_Ac6L · 2025-10-31

**Soundness:** 3
**Presentation:** 3
**Contribution:** 3
**Rating:** 6
**Confidence:** 2

**Summary:**

This article focuses on embedding causal knowledge in the form of a Directed Acyclic Graph (DAG) into a transformer architecture. The resulting model is designed to perform causal inference tasks. To achieve this, the authors introduce a causal transformer that enables this integration. They also implement a fully connected neural network that respect DAG structure as an additional baseline. Using a toy example, the authors demonstrate the importance of having a causal structure and a model that respects this causal structure. The work is further validated through real-world experiments.

**Strengths:**

* The authors provide clear motivation and demonstrate it with a compelling example that compares predictive power and treatment effect estimation.
* They propose a novel modification of the attention mechanism for embedding DAGs into the model.
* They introduce a formulation for incorporating DAG structure into neural networks.
* They benchmark their approach against other causal inference methods.

**Weaknesses:**

* The architecture needs to be trained for each DAG on each dataset, which can limit the method from being applied to larger datasets.

* While the authors describe the ability to handle multidimensional embeddings as an advantage, the experiments are conducted on tabular datasets. It would be valuable to see whether the transformer architecture can excel in such settings.

* The authors quantitatively justify not using normalization; however, it is not clear whether the pros would outweigh the cons.

**Questions:**

* Is there a way to amortize this approach across different DAGs or datasets?
* Are you confident that the differences between CAT and CCFCN are not due to insufficient hyperparameter tuning?
* Is it possible to create synthetic datasets for these models that include high-dimensional data?

---

> ### Author Response · Authors · 2025-11-20
> **Response to Reviewer Ac6L Part 1/N**
>
> We thank the reviewer for taking the time to provide constructive feedback, and for acknowledging the novelty, quality of the benchmark and clarity of motivation and demonstration. Below we respond to each point in turn.
>
> **Weakness 1 and Question 1:** *The architecture needs to be trained for each DAG on each dataset, which can limit the method from being applied to larger datasets. + Is there a way to amortize this approach across different DAGs or datasets?*
>
> Firstly, we thank the reviewer for their thoughts here and in their subsequent question. This is a very exciting area of future work for us.
>
> We agree that a naive implementation is DAG-specific. There are two parts in the original submission which already partly address this, and we outline these below along with their natural extensions.
>
> 1. Order-agnostic training via isomorphic shuffling (Described in Appendix S7). The model is trained under random permutations that preserve graph topology - with adjacency $A \in \{0,1\}^{\vert Z \vert \times \vert Z \vert}$ and a random permutation $p$ over $\{1,\dots,\vert Z \vert\}$, we form the permutation matrix $P$ and apply $A' = P A P^T$. We simultaneously permute $X$ and targets so that variable slots and the mask remain aligned. This forces the parameters to implement the same causal constraint for any node ordering, which reduces sensitivity orr overfitting to specific topological graph index, and improves reuse of weights across isomorphic layouts. In our experiments, shuffling modestly worsened MSE and widened ATE SEs, so we disabled it for the final experiments, but it remains a valid path to amortize across permutations of the same DAG and may help in settings closer to autoregressive masking where order-agnostic training has been beneficial. It is interesting and reassuring that we followed a similar line of thought to the reviewer here.
>
> 2. Family-wise amortization across related DAGs. Beyond permutations, CaT can share most parameters across a family of DAGs in several practical ways:
>    - Supergraph training. One could train the model once with a supergraph mask that contains the union of edges seen across tasks, then deactivate edges per task at inference by zeroing the corresponding mask entries. This avoids full retraining while still preseving the necessary hard exclusions determined by the DAG used at inference time.
>    - (b) Mixture-of-experts over masks. Route tokens through a small set of experts with learned gates that are functions of the current mask. This selects computation paths without duplicating the entire network.
>    - (c) Soft DAG inference as an auxiliary objective - one could learn a continuous mask with an acyclicity penalty during pretraining, then project to a hard mask at deployment when a specific DAG is provided. CaT’s architecture still enforces the 'locality' once the mask is fixed.
>
> ---
>
> **Weakness 2:** *While the authors describe the ability to handle multidimensional embeddings as an advantage, the experiments are conducted on tabular datasets. It would be valuable to see whether the transformer architecture can excel in such settings.*
>
> We thank the reviewer for highlighting this. Indeed, the full potential of CaT is not evident until it is applied to data with vector-valued datasets. Nonetheless, we have already evaluated CaT on a real psychology application with multi-dimensional nodes, which is described in Section 4.2. Here, we use the full questionnaire responses for each psychological construct as the vector-dimensioned inputs, rather than “collapsing” the responses down to a simple mean or weighted average, as is most common in the empirical literature. We then apply the same graph as was used in a parallel study with the same data, and compare the results against a doubly-robust method, Targeted Learning, operating on the univariate versions of the constructs. We find consistent results, which is promising because the Targeted Learning approach is generally superior owing to its doubly-robust properties. This demonstrates that CaT handles vector-valued variables in practice.
>
>
> More generally, outside of empirical/applied sciences, a practical reason most public benchmarks appear tabular is that there are very few standardized datasets that provide a vetted DAG over variables that themselves are multi-dimensional embeddings. To our knowledge, benchmarks with ground-truth causal structure across nodes that are vectors are rare. That is why we included the psychology study, where a prior DAG exists from other litearture and each construct is naturally multi-item.

---

> > ### Author Response · Authors · 2025-11-20
> > **Response to Reviewer Ac6L Part 2/2**
> >
> > ---
> >
> > **Weakness 3:** *The authors quantitatively justify not using normalization; however, it is not clear whether the pros would outweigh the cons.*
> >
> > We thank the reviewer for highlighting this, and indeed our wording was not ideal.
> >
> > To clarify our position: while LayerNorm can be applied without violating the structural causal constraints when used carefully (specifically, when applied only to the embedding dimension $C$ rather than across variables), we opted to exclude it due to concerns about calibration and effect size estimation, particularly under interventions.
> >
> > The reviewer is correct that standard LayerNorm applied to the embedding dimension does not create spurious dependencies between causally unconnected variables. Each variable's embedding is normalized independently, preserving the node-wise statistical independence enforced by our DAG constraints.
> >
> > In contrast, LayerNorm alters the scale and location of representations in a data-dependent manner. Consider two key scenarios:
> >
> > 1. **Intervention calibration.** When performing interventions via our recursive inference algorithm (Algorithm 1), we set intervention variables to specific values (e.g., $do(D=1)$ vs. $do(D=0)$). LayerNorm would transform these intervention values based on sample-dependent statistics, potentially distorting the intervention value. For instance, setting $D=1$ for all samples would result in different normalized values depending on the other features in each sample's embedding.
> > 2. **Effect size interpretation.** Average Treatment Effects (ATEs) are defined on the original scale of the outcome variable. LayerNorm applies affine transformations that vary across samples and layers, making it difficult to recover effect sizes on the interpretable original scale. While the relative ordering of effects might be preserved, the magnitude, which is critical for causal inference, becomes confounded with the normalization statistics.
> >
> > We acknowledge there is a trade-off: with LayerNorm, we would likely have better training stability and potentially faster convergence. For our purposes, where precise effect size estimation is paramount, we chose to omit LayerNorm. However, it may be possible to “train out” these effects, and we leave this to further work.
> >
> > We have revised the end of Section 3.2 accordingly.
> >
> > ---
> >
> > **Question 2:** *Are you confident that the differences between CaT and CFCN are not due to insufficient hyperparameter tuning?*
> >
> > Indeed it is quite possible that with hyperparameter tuning (which we did not undertake) we would likely see closer performance between the two models. Nonetheless, overall we see similar performance across tasks for both models in cases where the graphs are correctly specified. This is likely because CFCN is limited to univariate input variables, and the learning problem is thus less complex.
> >
> > ---
> >
> > **Question 3:** *Is it possible to create synthetic datasets for these models that include high-dimensional data?*
> >
> > We thank the reviewer for the suggestion and agree that it would absolutely be possible to create synthetic datasets which include multivariate data. As mentioned above, we already validate this scenario with the real-world application to psychological data, but we acknowledge that a more in-depth exploration of performance could be made with synthetic data (which, for example, might follow an extended DGP to those used in other benchmarks).

---

### Official Review · Reviewer_E5kE · 2025-11-04

**Soundness:** 2
**Presentation:** 3
**Contribution:** 2
**Rating:** 4
**Confidence:** 4

**Summary:**

The paper introduces Causal Transformers (CaTs), which incorporates causal graphs as masks inside the attention mechanism of transformers, which improves their performance in terms of robustness and interpretability. CaTs are capable of estimating causal queries such as treatment effects, and they are shown experimentally to outperform methods that do not take causality into account, while being able to compete with methods that do.

**Strengths:**

1. Incorporating graphs into the attention mechanism ensures that the causal constraints encoded in the graphs (and therefore the resulting causal inferences) are guaranteed to hold. This is a strong advantage over methods that attempt to achieve causality through regularization. The adjustments such as the omission of layer norm are done carefully to preserve this causal integrity.

2. Assumptions are stated clearly.

3. The provided experimental results seem promising.

**Weaknesses:**

4. The contributions of the paper are largely empirical and leave much to be desired from the theoretical aspects of the models. Most notably, I believe it would be a core contribution to include a theorem that guarantees that causal constraints (such as those from Causal Bayesian Networks) are correctly enforced in the CaT architecture, and therefore, the estimated queries will be correct. Another important contribution would be to support the claim that causal models like CaT can resolve issues related to covariate shift, so it would be fundamental to include a theorem that shows what kinds of guarantees CaT can provide across domains (perhaps using causal transportability theory).

5. The training details of CaT are somewhat unclear. In particular, it would be great to see the exact details of how the loss function is implemented, and how causality plays a role in it.

6. It is unclear what kinds of interventional queries CaTs are capable of estimating, and how each of those queries are particularly calculated through the CaT architecture. It seems that one weakness is that CaTs are not capable of estimating ALL interventional queries (perhaps only the causal effect of treatment on target).

7. Notation is not introduced properly. It is not clear what $B$, $Z$ and $C$ are.

**Questions:**

8. What makes CFCN an interesting baseline, as opposed to just another method developed by the paper? It seems like an oddly specific inclusion to the paper.

9. The paper claims in Sec. 4.2 that it is not expected for CaT and CFCN to excel compared to models specifically tuned for causal inference tasks. I would have listed this as a weakness, but I believe that perhaps one strength of these models is their generative capabilities across many different queries. Could the authors confirm if this is the case (e.g., could CaTs produce novel samples of both the observational and interventional data distributions)? If so, how does this model compare to common approaches used in the causal generative modeling literature such as [1][2][3]?

[1] Kocaoglu, et al., “CausalGAN: Learning Causal Implicit Generative Models with Adversarial Training” (2017).

[2] Xia, et al., “The Causal-Neural Connection: Expressiveness, Learnability, and Inference” (2021).

[3] Saremi, “Neural Network Parameter-optimization of Gaussian Pre-marginalized Directed Acyclic Graphs” (2023).

---

> ### Author Response · Authors · 2025-11-20
> **Response to Reviewer E5kE Part 1/N**
>
> We thank the reviewer for their consideration, time, and constructive feedback, as well as for their kind words regarding the “strong advantage” that CaT holds over regularization-based methods. Below we respond point by point.
>
> **Point 4:** *The contributions of the paper are largely empirical and leave much to be desired from the theoretical aspects of the models. Most notably, I believe it would be a core contribution to include a theorem that guarantees that causal constraints (such as those from Causal Bayesian Networks) are correctly enforced in the CaT architecture, and therefore, the estimated queries will be correct. Another important contribution would be to support the claim that causal models like CaT can resolve issues related to covariate shift, so it would be fundamental to include a theorem that shows what kinds of guarantees CaT can provide across domains (perhaps using causal transportability theory).*
>
> We have added a new theoretical section in the supplementary material. Theorem 1 shows that, under our architectural assumptions, CaT (defines a Causal Bayesian Network which) has conditional factorisation that matches that implied by the supplied DAG and that interventions implemented through the masking and embedding mechanism coincide with the standard g–formula (Robins, 1986). Theorem 2 and Corollary 1 formalise the covariate shift and transportability claims: if the causal mechanisms are invariant across domains, interventional queries (such as ATE) are invariant across domains, and predictions for a target node are insensitive to shifts in variables outside its ancestor set.
>
> We have also pasted these new contributions to the end of this set of responses, for convenience.
>
> ---
>
> **Point 5:** *The training details of CaT are somewhat unclear. In particular, it would be great to see the exact details of how the loss function is implemented, and how causality plays a role in it.*
>
> A strength of CaT (as the reviewer also noted) is that there are no special losses or regularizers. We simply constrain the architecture to the DAG and train each endogenous variable to predict its observed value (as such, exogenous nodes are inputs only and receive no loss):
>
> For any endogenous node $z$: if $z$ is continuous, we use mean squared error on its output head; if binary, Bernoulli cross-entropy; if categorical, softmax cross-entropy; if a vector embedding, MSE on that vector. The experimenter is thus free to implement the loss they wish, understanding that the predictions are constrained according to the assumed DAG. The total loss is then the sum over node-specific losses, optionally weighted if needed for scale balancing. For optimization one can use standard choices like Adam or AdamW -  nothing causal-inference-specific is required.
>
> We have clarified this in Section 3.4.
>
> ---
>
> **Point 6:** *It is unclear what kinds of interventional queries CaTs are capable of estimating, and how each of those queries are particularly calculated through the CaT architecture. It seems that one weakness is that CaTs are not capable of estimating ALL interventional queries (perhaps only the causal effect of treatment on target).*
>
> CaT supports all causal effect queries identifiable from the DAG via standard identification, operationalized as follows: set intervened variables to their intervention values, then recursively update descendants in topological order by re-evaluating the corresponding node heads with parents fixed to their current values. This yields total effects, joint interventions, and path-specific effects if the mask is reduced to the desired active paths (so this includes, for example, conditional average (CATE). average (ATE) or quantile effects by extension). CaT does not identify queries that are unidentifiable in the given DAG (and we say so plainly, and on a similar note, there is currently no latent inference). We have clarified this in Section 3.5.
>
> ---
> **Point 7:** *Notation is not introduced properly. It is not clear what $B$, $Z$ and $C$ are.*
>
> We apologise for this oversight. $B$, $Z$, and $C$ are batch size, the number of nodes in the DAG, and the raw variable feature dimension, respectively. We have updated this in Section 3 and include a full list of terms/definitions in the supp. mat.
>
> **Question 8:** *What makes CFCN an interesting baseline, as opposed to just another method developed by the paper? It seems like an oddly specific inclusion to the paper.*
>
> We thank the reviewer for the question. CFCN isolates the value of structural masking but without attention. Using the same DAG mask with layered identity handling, it serves as an architectural control to show that benefits are not unique to the attention mechanism of the transformer. Although approximate, the differences between CaT and CFCN quantify the specific contribution of attention under the same causal constraint. We have clarified this at the start of Section 4.

---

> > ### Author Response · Authors · 2025-11-20
> > **Response to Reviewer E5kE Part 2/N**
> >
> > **Question 9:** *The paper claims in Sec. 4.2 that it is not expected for CaT and CFCN to excel compared to models specifically tuned for causal inference tasks. I would have listed this as a weakness, but I believe that perhaps one strength of these models is their generative capabilities across many different queries. Could the authors confirm if this is the case (e.g., could CaTs produce novel samples of both the observational and interventional data distributions)? If so, how does this model compare to common approaches used in the causal generative modelling literature such as [1][2][3]*
> >
> > We thank the reviewer for bringing up a very interesting point. Indeed, it is possible to use CaTs in a generative mode, even though they are not inherently generative models. Rather, CaT is a conditional predictor that could be used generatively by specifying noise models per node and sampling exogenous noise, then pushing samples forward through the DAG. This mode of operation is not dissimilar to its standard function in the presence of an intervention: if one manipulates the values of certain inputs and pushes them through CaT, one can observe a distribution over the outputs. This enables sampling from a mixture of observational and interventional distributions. Unlike adversarial methods such as CausalGAN, CaT’s guarantees come from hard architectural masking rather than adversarial training or specific parametric factorization.
> >
> > Relative to the other articles, our positioning is as follows. CausalGAN [1] is an implicit generative model trained adversarially to match observational and interventional distributions over high dimensional outputs such as images, with a generator architecture constrained by a causal graph. In contrast, CaT is not an adversarial generator but more of a mechanism learner: it parametrizes the conditional distributions of a structural causal model over arbitrary scalar or embedding-valued nodes and then supports sampling and interventional queries through the DAG. In this sense CaT is closer in spirit to the neural causal models of Xia et al. [2], which emphasize neural parametrizations of structural equations and identifiability limits, and to the parameter optimization of DAG based Gaussian models in Saremi [3]. Our contribution is complementary to these: we provide a concrete transformer-style architecture with hard architectural masking that (i) enforces the graphical constraints by design, (ii) scales to multimodal, high dimensional embeddings, and (iii) can be used both for causal effect estimation and for generative sampling once node-wise noise models are specified.
> >
> > ---
> >
> > We now provide the theoretical proofs that the reviewer request (and which have been added to the last section in the supp. mat.):
> > ### Architectural assumptions and causal semantics
> >
> > #### Review of terminology and notation
> >
> > The tensor $\mathbf{X} \in \mathbb{R}^{B \times |Z| \times C}$ contains, for each mini-batch element, the $C$-dimensional raw features for every node in the DAG. The slice $\mathbf{X}_i = \mathbf{X}[:, i, :]$ is the raw input for node $i \in Z$.  Embeddings are produced by $|Z|$ independent linear maps to give $\mathbf{X}_E \in \mathbb{R}^{B \times |Z| \times d_E}$  where $d_E \geq C$ and $d_E > 1$.
> >
> > The learnable query embedding is $\mathbf{Y} \in \mathbb{R}^{|Z| \times d_E}$. The topologically sorted adjacency is $\mathbf{A} \in \{0,1\}^{|Z|\times |Z|}$ with $\mathbf{A}_{ji} = 1$ if and only if $j \in \mathrm{pa}(i)$.
> >
> > #### Cross-attention structure
> > Unlike standard self-attention, CaT employs causally-masked cross-attention (CMCA) where queries $\mathbf{Q} = \mathbf{Y}\mathbf{W}_Q + \mathbf{b}_Q$ derive from the learnable embedding $\mathbf{Y}$ in the first block (then from previous block outputs), while keys $\mathbf{K} = \mathbf{X}_E\mathbf{W}_K + \mathbf{b}_K$ and values $\mathbf{V} = \mathbf{X}_E\mathbf{W}_V + \mathbf{b}_V$ always derive from the embedded input $\mathbf{X}_E$. This enables iterative refinement by extracting parental information at each layer.
> >
> > **Assumption 1 (Hard parental mask).** At every block, attention from token $i$ can only read from its parents $\mathrm{pa}(i)$. This is enforced through
> > $
> > \mathbf{O} = \text{softmax}\left(\mathbf{A}^\top \circ \frac{\mathbf{Q}\mathbf{K}^\top}{\sqrt{h_s}}\right) \cdot \mathbf{V}$
> >
> > where
> >
> > $(\mathbf{A}^\top)_{ij} = 1$
> >
> > if and only if $j \in \mathrm{pa}(i)$.
> >
> > Since $\mathbf{A}$ has zeros on the diagonal (no self-loops), $(\mathbf{A}^\top)_{ii} = 0$, preventing self-attention.
> >
> > Thus attention weights from any $S \subseteq Z \setminus \mathrm{pa}(i)$ to $i$ are zero.
> >
> > **Assumption 2 (Per-node embedding).**
> >
> > The embedding stage uses $|Z|$ independent linear layers, so the embedded token $\mathbf{X}_{E,i}$ depends only on $\mathbf{X}_i$. This preserves causal independence at the input level and enables heterogeneous feature dimensions across nodes.

---

> > > ### Author Response · Authors · 2025-11-20
> > > **Response to Reviewer E5kE Part 3/N**
> > >
> > > **Assumption 3 (Tokenwise operations).** Residual connections and the positionwise feed-forward $FF(\cdot)$ act independently across tokens. Each CBlock preserves this structure:
> > >
> > > $\mathbf{O}_{\text{CMCA}}^{r,'} = \text{CMCA}(\mathbf{O}_{\text{Block}}^{r-1}, \mathbf{X}_E) + \mathbf{O}_{\text{Block}}^{r-1}$
> > >
> > > $\mathbf{O}_{\text{Block}}^r = FF(\mathbf{O}_{\text{CMCA}}^{r,'}) + \mathbf{O}_{\text{CMCA}}^{r,'}$
> > >
> > > where $\mathbf{O}_{\text{Block}}^0 = \mathbf{Y}$ and batch normalization is optional.
> > >
> > > No layer normalization is used as it would rescale potentially calibrated interventions.
> > >
> > > ### Layerwise dependence and local Markov property
> > >
> > > Let $h_i^{(\ell)}$ denote token $i$'s hidden state after block $\ell$; by construction $h_i^{(L)}$ feeds the prediction head for node $i$.
> > >
> > > **Proposition 1 (Parental sufficiency).** Under Assumptions 1–3, for any block $\ell \geq 1$ and node $i \in Z$, there exists a measurable function $F_i^{(\ell)}$ such that
> > > $h_i^{(\ell)} = F_i^{(\ell)}\big(\mathbf{X}_{\mathrm{pa}(i)}\big)$
> > >
> > > Consequently, for any $S \subseteq Z \setminus \mathrm{pa}(i)$,
> > > $h_i^{(\ell)} \perp\!\!\!\perp \mathbf{X}_S \mid \mathbf{X}_{\mathrm{pa}(i)} $
> > >
> > > **Intuition.** Token $i$ aggregates only from its parents through causally-masked cross-attention and cannot attend to itself due to the zero diagonal in $\mathbf{A}$. Since $\mathbf{X}_E$ is fed at every layer, each block allows the evolving query (from $\mathbf{O}_{\text{Block}}^{r-1}$) to extract relevant parental information (keys/values from $\mathbf{X}_E$).
> > >
> > > The residual connections maintain and refine token $i$'s representation across layers without requiring self-attention. The hidden state at $i$ thus depends only on parental inputs $\mathbf{X}_{\mathrm{pa}(i)}$.
> > >
> > > **Remark 1 (Strict causal dependency through masking).** The architecture enforces strict causal dependency: token $i$ never directly accesses $\mathbf{X}_i$ through attention. The DAG mask with zero diagonal ensures $h_i^{(\ell)}$ is constructed purely from parental information.
> > >
> > > Let $\widehat{\mathbf{X}}_i$ be the output of the final linear prediction head for node $i$, computed from $h_i^{(L)} = \mathbf{O}_{\text{Block},i}^L$ after $L$ CBlocks.
> > >
> > >
> > > **Lemma 1 (Local Markov property).** Under Assumptions 1–3, with the final linear layer acting independently per token,
> > >
> > > $\widehat{\mathbf{X}}_i = g_i\big(\mathbf{O}_{\text{Block},i}^L\big) = g_i\big(F_i^{(L)}(\mathbf{X}_{\mathrm{pa}(i)})\big).$
> > >
> > > Since the mask structurally prevents self-attention, the prediction is necessarily a function of $\mathbf{X}_{\mathrm{pa}(i)}$ alone, yielding
> > >
> > > $p_{\text{CaT}}\big(x_i \mid x_{Z\setminus\{i\}}\big) = p_{\text{CaT}}\big(x_i \mid x_{\mathrm{pa}(i)}\big).$
> > >
> > > **Intuition.** The final prediction head reads token $i$'s representation after all CBlocks. Since the DAG mask has zero diagonal, token $i$ never accesses $\mathbf{X}_i$ during forward passes. This architectural constraint enforces the local Markov property of a Causal Bayesian Network with respect to $\mathcal{G}$.
> > >
> > >
> > > **Theorem 1 (Structural soundness and DAG factorization).** Under Assumptions 1–3, the joint predictive distribution defined by CaT factorizes according to the DAG $\mathcal{G}$:
> > >
> > > \[
> > > p_{\text{CaT}}(x_1,\ldots,x_{|Z|}) = \prod_{i \in Z} p_{\text{CaT}}\!\big(x_i \mid x_{\mathrm{pa}(i)}\big).
> > > \]
> > >
> > > In particular, CaT implements a Causal Bayesian Network over $Z$ whose conditional distributions respect the parental sets in $\mathcal{G}$.
> > >
> > > **Intuition.** The CMCA mask guarantees that each node's representation depends only on its parents. The parallel attention computation thus yields a model that satisfies the local Markov property for all nodes. The standard result from graphical models then implies the global factorization with respect to $\mathcal{G}$.
> > >
> > > ### Identifiable interventions via masked attention
> > >
> > > We now present how CaT implements the truncated factorization (g-formula) used in causal inference (Robins, 1986).
> > >
> > > **Definition 1.** For an intervention set $\mathcal{I}=\{(\mu,z): z \in S\}$ with $S \subseteq Z$, CaT implements $do(X_S=\mu)$ by:
> > >
> > > 1. Replacing embeddings for intervened nodes: $\mathbf{X}_{E,j}^{do} = \text{Embed}_j(\mu)$ for $j \in S$.
> > > 2. Processing $\mathbf{X}_E^{do}$ through all CBlocks with unchanged CMCA masks.
> > > 3. Reading outputs where intervened nodes retain their set values:
> > >
> > > \[
> > >    X^{do}_j =
> > >    \begin{cases}
> > >    \mu, & j \in S,\\[2pt]
> > >    \widehat{X}_j\!\big(\mathbf{O}_{\text{Block},j}^L\big), & j \notin S.
> > >    \end{cases}
> > > \]
> > >
> > > Non-intervened nodes automatically incorporate upstream interventions since their representations depend only on (possibly intervened) parents.

---

> > > > ### Author Response · Authors · 2025-11-20
> > > > **Response to Reviewer E5kE Part 4/4**
> > > >
> > > > (we apologies for the failed math rendering in the previous and current responses, we were unable to fix these and recommend the updated PDF for reference).
> > > >
> > > > **Proposition 2 (Truncated product via CMCA).** Under Theorem 1, the CMCA architecture with Definition 1 directly implements the g-formula (Robins, 1986)
> > > > $p_{\text{CaT}}\!\big(x_{Z} \mid do(x_S)\big) = \prod_{j \in Z \setminus S} p_{\text{CaT}}\!\big(x_j \mid x_{\mathrm{pa}(j)}\big)\Big|_{x_{\mathrm{pa}(j)} \leftarrow x^{do}_{\mathrm{pa}(j)}}.$
> > > >
> > > > Treatment effects like $\boldsymbol{\tau} := \mathbb{E}\big[Y \mid do(D{=}\mu_1)\big] - \mathbb{E}\big[Y \mid do(D{=}\mu_0)\big]$ are therefore computed by forward passes with different intervention values.
> > > >
> > > > **Intuition.** Interventions work naturally in CaT: setting $\mathbf{X}_{E,j}$ to intervention values affects only downstream nodes (children of $j$), since node $j$ itself cannot self-attend. The mask ensures each node reads only from parents, so intervention effects propagate causally downstream in a single forward pass. The resulting interventional distribution coincides with the truncated product of a Causal Bayesian Network over $\mathcal{G}$.
> > > >
> > > > ### Covariate shift, structural robustness, and transportability
> > > >
> > > > We now formalize how CaT behaves under covariate shift and connect this to standard assumptions in causal transportability. The key result is a *structural robustness* property: assuming that the DAG is correct and the causal mechanisms are invariant, CaT's predictions do not change when covariates that are not causal ancestors of the target shift between domains. The usual transportability of interventional queries then follows as a consequence.
> > > >
> > > > Consider two domains (or environments) $e \in \{s,t\}$ with the same DAG $\mathcal{G}$ and the same structural conditionals
> > > >
> > > > $p^{(e)}\big(x_i \mid x_{\mathrm{pa}(i)}\big) = p\big(x_i \mid x_{\mathrm{pa}(i)}\big)
> > > > \quad\text{for all } i \in Z$
> > > >
> > > > but possibly different observational distributions $p^{(e)}(x_Z)$ due to different exogenous noise or covariate distributions.
> > > >
> > > > **Assumption (Consistent estimation of conditional distributions).**
> > > >
> > > > CaT is trained in domain $s$ with sufficient capacity and data so that for all $i \in Z$,
> > > >
> > > > $p_{\text{CaT}}^{(s)}\big(x_i \mid x_{\mathrm{pa}(i)}\big) \xrightarrow[]{N\to\infty} p\big(x_i \mid x_{\mathrm{pa}(i)}\big).$
> > > >
> > > > The same architecture is then applied in domain $t$ without retraining.
> > > >
> > > > **Theorem (Structural robustness under covariate shift).**
> > > >
> > > > Let $Y$ be a target node, and suppose domain shift affects only a subset $S \subseteq Z \setminus \mathrm{An}(Y)$ that contains no ancestors of $Y$. Under Assumptions (Mask), (Tokenwise), and (Consistency), for any fixed configuration of parental variables $x_{\mathrm{pa}(Y)}$, the CaT predictor for $Y$ is invariant across such covariate shift:
> > > >
> > > > $p_{\text{CaT}}^{(s)}\big(y \mid x_{\mathrm{pa}(Y)}\big)$
> > > >
> > > > $=p_{\text{CaT}}^{(t)}\big(y \mid x_{\mathrm{pa}(Y)}\big)
> > > > \quad\text{for all } y, x_{\mathrm{pa}(Y)}.$
> > > >
> > > > In other words, CaT is robust to covariate shift in variables that are not causal ancestors of $Y$.
> > > >
> > > > **Intuition.**
> > > > Due to the CMCA mask, the representation for $Y$ can only depend on its parents and their ancestors. Variables outside $\mathrm{An}(Y)$ never influence $Y$'s token representation. In contrast, non-causal transformers/networks are vulnerable to this: in a generic neural network, the representation of $Y$ is free to depend on any input variable, so non-ancestors can spuriously influence predictions and make them sensitive to distributional changes that do not alter the underlying mechanisms.
> > > >
> > > > **Corollary (Transportability of identified causal queries).**
> > > > Under Assumptions (Mask), (Tokenwise), and (Consistency), any causal query that is identified by the g-formula (Robins, 1986) in $\mathcal{G}$, such as the average treatment effect
> > > >
> > > > $\tau = \mathbb{E}\big[Y \mid do(D{=}\mu_1)\big] - \mathbb{E}\big[Y \mid do(D{=}\mu_0)\big],$
> > > >
> > > > has a CaT estimate that is invariant across domains $s$ and $t$ in the large-sample limit:
> > > >
> > > > $\tau_{\text{CaT}}^{(s)} \xrightarrow[]{N\to\infty} \tau
> > > > \quad\text{and}\quad
> > > > \tau_{\text{CaT}}^{(t)} \xrightarrow[]{N\to\infty} \tau.$
> > > >
> > > > Equivalently, as long as the mechanisms $p(x_i \mid x_{\mathrm{pa}(i)})$ are shared across domains, CaT's interventional estimates are transportable.

---

### Meta-Review · Area_Chair_MY9j · 2026-01-06

**Summary:**

The work proposes Causal Transformers (CaTs), which integrate causal graphs into the Transformer attention mechanism via structured masks, that the authors claim, lead to improved robustness and interpretability. By explicitly encoding causal structure, the paper shows that CaTs can estimate causal quantities such as treatment effects. Experimental results show that CaTs outperform approaches that ignore causality, while remaining competitive with methods specifically designed for causal inference.

The paper received 3 reviews with 1 reviewer recommending accept and the other 2 reviewers being borderline, slightly leaning towards reject. The major points of contention of the reviewers were as follows:

* Weak theoretical underpinnings. Reviewer E5kEmentions and I quote "it would be a core contribution to include a theorem that guarantees that causal constraints (such as those from Causal Bayesian Networks) are correctly enforced in the CaT architecture, and therefore, the estimated queries will be correct. " ANother point is "to show that causal models like CaT can resolve issues related to covariate shift, so it would be fundamental to include a theorem that shows what kinds of guarantees CaT can provide across domains"

* Scaling concerns. A reviewer points out that the architecture is DAG specific and since the DAGs can get complex for large data sets, training the architecture can get tricky.

**Reviewer Concerns:**

I think the authors provided a very strong rebuttal and clarified most of the points of the reviewers. The authors provided a theorem for the structural robustness under covariate shift and the structural soundness with respect to the proposed architecture. I read through the proofs (albeit not in too much detail I must confess) and they seem correct to me.

Regarding the scaling question, I feel that the authors did answer it although not in a very convincing manner. I do not hold this against the authors since scaling in causality is a difficut question and I do not expect all the questions to be tackled in a single paper.

**Reviewer Scores:**

Based on the rebuttals I do expect both reviewers to raise their score (atleast to a 6).

I feel that the paper is sound and presents a nice way to bridge causality with transformers thereby paving the way for incorporating causal structure in the currently most studied ML model. I have seen other papers try the same but this work is till now the most structured and principled approach. I thus recommend acceptance and hope that the authors take care of all outstanding comments of the authors in the final version of the paper.

---

### Decision · Program_Chairs · 2026-01-26

Accept (Poster)